# Diagnostic and commensal *Staphylococcus pseudintermedius* genomes reveal niche adaptation through parallel selection of defense mechanisms

Sanjam S. Sawhney [1,8], Rhiannon C. Vargas [1,8], Meghan A. Wallace[2], Carol E. Muenks[2], Brian V. Lubbers[3], Stephanie A. Fritz[4], Carey-Ann D. Burnham[2,4,5,6] ✉ & Gautam Dantas [1,2,4,6,7] ✉

*Staphylococcus pseudintermedius* is historically understood as a prevalent commensal and pathogen of dogs, though modern clinical diagnostics reveal an expanded host-range that includes humans. It remains unclear whether differentiation across *S. pseudintermedius* populations is driven primarily by niche-type or host-species. We sequenced 501 diagnostic and commensal isolates from a hospital, veterinary diagnostic laboratory, and within households in the American Midwest, and performed a comparative genomics investigation contrasting human diagnostic, animal diagnostic, human colonizing, pet colonizing, and household-surface *S. pseudintermedius* isolates. Though indistinguishable by core and accessory gene architecture, diagnostic isolates harbor more encoded and phenotypic resistance, whereas colonizing and surface isolates harbor similar CRISPR defense systems likely reflective of common household phage exposures. Furthermore, household isolates that persist through anti-staphylococcal decolonization report elevated rates of base-changing mutations in – and parallel evolution of – defense genes, as well as reductions in oxacillin and trimethoprim-sulfamethoxazole susceptibility. Together we report parallel niche-specific bolstering of *S. pseudintermedius* defense mechanisms through gene acquisition or mutation.

The *Staphylococcus intermedius* group (SIG) consists of closely related coagulase-positive staphylococcal (CoPS) species *S. pseudintermedius*, *S. intermedius*, *S. delphini*, *S. cornubiensis*, and *S. ursi*, which together have long been characterized for their wide host range of domesticated and wild animal species[1,2]. *S. pseudintermedius*, the most frequently recovered member from clinical specimens, exists commensally on most dogs yet is also the most common etiologic agent of canine pyoderma, ear, postoperative, and urinary tract infections[1–4]. Biochemically, SIG species appear remarkably similar to the archetype human CoPS pathogen, *S. aureus*, likely contributing to a historic

[1]The Edison Family Center for Genome Sciences and Systems Biology, Washington University School of Medicine, St. Louis, MO, USA. [2]Department of Pathology and Immunology, Division of Laboratory and Genomic Medicine, Washington University School of Medicine, St. Louis, MO, USA. [3]Department of Clinical Sciences, Kansas State University, Manhattan, KS, USA. [4]Department of Pediatrics, Washington University School of Medicine, St. Louis, MO, USA. [5]Department of Medicine, Washington University School of Medicine, St. Louis, MO, USA. [6]Department of Molecular Microbiology, Washington University School of Medicine, St. Louis, MO, USA. [7]Department of Biomedical Engineering, Washington University in St. Louis, St. Louis, MO, USA. [8]These authors contributed equally: Sanjam S. Sawhney, Rhiannon C. Vargas. ✉e-mail: cburnham@wustl.edu; dantas@wustl.edu

underreporting of SIG as a contributor to human infection[5–9]. Indeed, recent advancements in laboratory typing methods, including repetitive sequence PCR, MALDI-TOF mass spectrometry, and whole-genome sequencing (WGS), have revealed *S. pseudintermedius* as an opportunistic pathogen found in human wounds, as well as respiratory, skin and soft tissue, and bloodstream infections[9–12]. Though the earliest reports of human *S. pseudintermedius* infections were linked with exposure to companion animals[8,13], larger retrospective studies have reported a history of animal contact in just 6–10% of human *S. pseudintermedius* infections[9,11], indicating the clinical context of *S. pseudintermedius* disease in humans extends beyond direct zoonotic transmission. Prevalence of methicillin resistance[14]—which is very closely tied to multidrug resistance—among *S. pseudintermedius* canine infections has increased dramatically from <5% in 2001 to 20–45% by 2017[14–17]. Though methicillin-resistant *S. pseudintermedius* (MRSP) prevalence among human infections remains largely unknown, early reports indicate a congruence with methicillin-resistant *Staphylococcus aureus* (MRSA) rates among Western populations[2,9,12,18,19], signaling a burgeoning, serious, public health concern.

WGS has enabled comparative genomics studies that have provided foundational knowledge of *S. pseudintermedius* populations and pathologies[20,21]. Similar to *S. aureus*, *S. pseudintermedius* has an open pangenome; most accessory genes are lineage- or strain-specific[20–23], and antibiotic resistance gene (ARG) content and virulence factor repertoires vary by multilocus sequence type (MLST)[20,24]. Additional studies have characterized *S. pseudintermedius* genome content by geographic isolation[21], host-species for clinical isolates[22], niche-type for veterinary isolates[20,25], and *mecA* presence[26]. These reports have identified circulation of phylogenetically-diverse lineages across companion animals[21] as well as differences in genome size between MRSP and methicillin-susceptible *Staphylococcus pseudintermedius* (MSSP)[25,26]. However, these efforts were unable to identify genes that tie diagnostic *S. pseudintermedius* populations to canine or human hosts[22], nor did they observe differences in virulence factor repertoire between clinical and colonizing veterinary isolates, blurring preconceptions of host-species boundaries for diagnostic isolates and suggesting that commensal veterinary *S. pseudintermedius* have pathogenic potential[25,27]. As there have been no studies contrasting isolates across both host-species (human vs. companion animals) and niche-type (diagnostic vs. colonizing), the relative influence of these two modalities on *S. pseudintermedius* population genomics remains unknown. Incorporation of cohorts from multiple host-species and niche-types is needed to determine the major contributing factors towards differentiation among *S. pseudintermedius* populations, and how such differentiation impacts both spread between companion animals and humans, as well as transitions between opportunistic pathogen and commensal lifestyles.

Households present individualized case studies within which *S. pseudintermedius* strain persistence, transmission, and replacement within a microenvironment can be observed. Prior studies reveal that humans and companion animals in a shared household have the potential to introduce *S. pseudintermedius* from outside sources, circulate strains amongst household members, and contribute to host reinfection following clearance of prior MRSP infection[28–31]. In addition, a 2011 observational study investigating the prevalence of MRSP in households with an index MRSP-positive companion animal indicated that environmental contamination was also widespread, finding sleeping, feeding, and lounging areas to also harbor *S. pseudintermedius*[28]. Further, a longitudinal study in the same year found that MRSP can be identified on these household surfaces without ongoing MRSP colonization or infection of a human or companion animal, indicating such abiotic surfaces may act as reservoirs that enable *S. pseudintermedius* persistence[32]. Recent investigations into *S. pseudintermedius* in households have supported these findings but suggest sampling from multiple sites over time is needed to better understand within-host diversity and within-household transmission of S. *pseudintermedius*[29]. Investigating

the propagation of *S. pseudintermedius* in the shared biotic and abiotic environment is a key element to understanding *S. pseudintermedius* as an opportunistic, multi-host pathogen[33].

In this study, we unite three cohorts of SIG isolates captured at a major tertiary care medical center and veterinary diagnostic laboratory, both in the American Midwest, and the Household Observation of MRSA in the Environment 2 (HOME2) and Staph Household Intervention for Eradication (SHINE) studies[34,35] that longitudinally sampled inhabitants and surfaces within households before, during, and after MRSA decolonization efforts in the St. Louis, Missouri, USA, metropolitan area. From these collection sites we have acquired, sequenced, and performed antibiotic susceptibility testing (AST) on over 500 human diagnostic, animal diagnostic, and household (human and pet colonizing, and environmental) isolates to investigate genotypic and phenotypic differences across host-species and niche-type.

## Results

### Isolate collection and characterization from three unique cohorts

Given the history of misidentification of SIG genomes amidst coagulase-positive staphylococci[8,10], our first aim was to validate MALDI-TOF MS calls against whole-genome sequencing. Towards this, we collected all 572 isolates reported as "*Staphylococcus intermedius/pseudintermedius*" (n = 565) or "*S. delphini*" (n = 7) by MALDI-TOF MS at the Barnes-Jewish clinical microbiology laboratory between December 2011 and July 2019. These isolates were originally captured at the Barnes-Jewish Hospital in St. Louis, MO, USA (n = 181), Kansas State University Veterinary Diagnostic Laboratory in Manhattan, KS, USA (n = 100), and within households in the St. Louis metropolitan area (n = 290) (Supplementary Fig. 1, Supplementary Data 1, 2). After Illumina sequencing and filtering by genome quality, we retained 501 isolate assemblies (n = 163 human diagnostic isolates, n = 94 veterinary diagnostic isolates, and n = 244 household isolates) that met our inclusion criteria (Supplementary Data 4). We retained all isolates called "*S. delphini*" (n = 7) by MALDI-TOF MS and most called "*S. intermedius/pseudintermedius*" (n = 494) after filtering.

### WGS validates MALDI-TOF MS as a precise tool for SIG identification

For species identification of the 501 isolates, we queried NCBI and downloaded 70 high-quality assemblies encompassing all SIG species. With these reference assemblies, we performed an all-against-all whole genome alignment using FastANI[36], reporting average nucleotide identity (ANI) and percent total alignment between each isolate comparison (Supplementary Fig. 2A). As expected, all reference assemblies clustered with other reference assemblies of the same species. None of the 501 undesignated isolates reported ≥95% ANI with either the *S. intermedius* or the *S. ursi* reference clusters. Eight of the 501 isolates were present within the *S. delphini* reference cluster (≥95% ANI) and the remaining (n = 493) were found among the *S. pseudintermedius* reference cluster; all isolates only had ≥95% ANI with a single reference species cluster. All seven isolates called by MALDI TOF MS as "*S. delphini*" (Supplementary Fig. 2B) were among the eight isolates mixed into the *S. delphini* reference cluster. The eighth was the only isolate among the 501 assessed to have been erroneously typed, miscalled as "*S. intermedius/pseudintermedius*." Thus, MALDI-TOF reports 99.80% and 100% precision when calling an isolate as "*S. intermedius/pseudintermedius*" (493/494) and "*S. delphini*" (7/7), respectively. Given its predominance within our cohort, we centered the remainder of our investigation on *S. pseudintermedius*.

### Niche adaptation is not apparent via global encoded gene architecture

Via in silico multilocus sequence typing (MLST), we find both a high level of diversity within our cohort, reflecting findings from

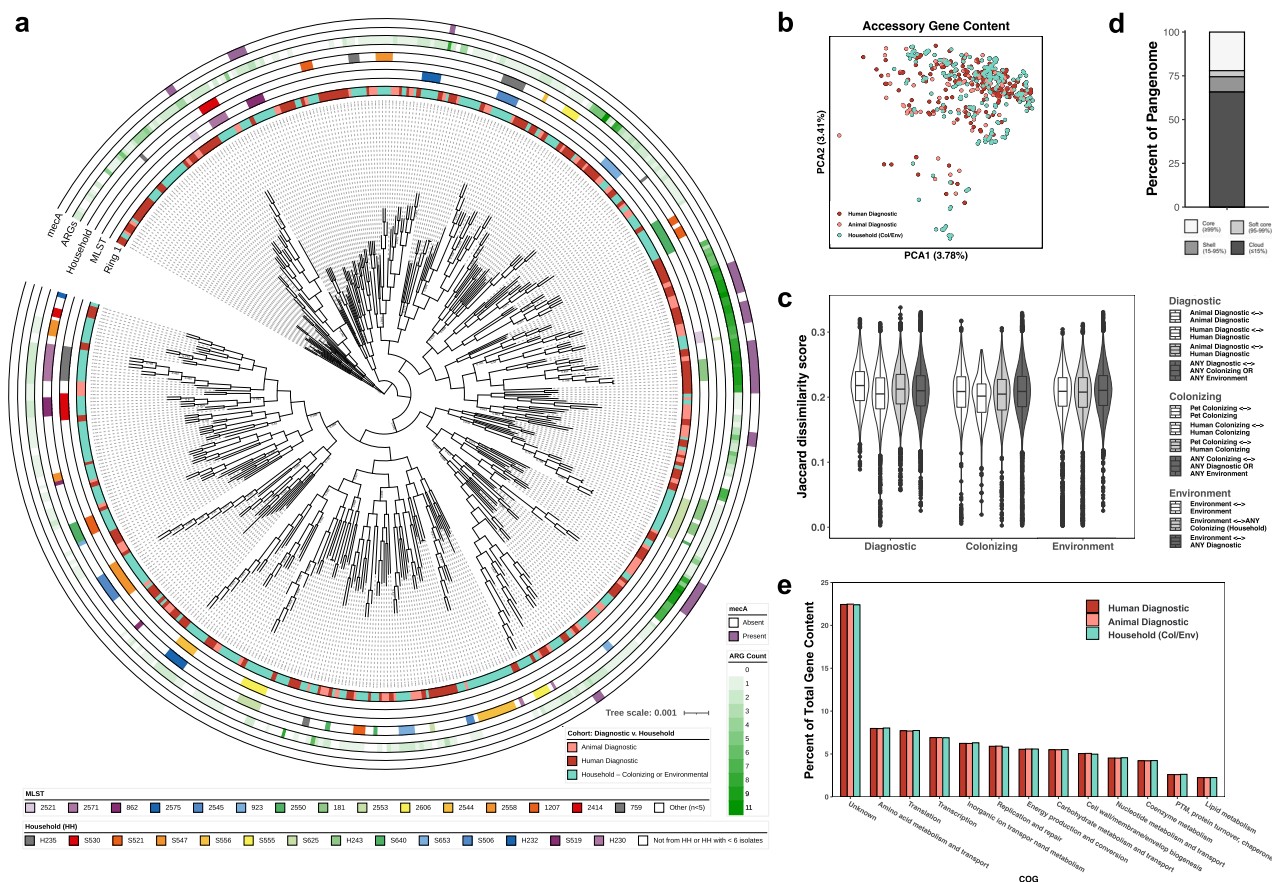

**Fig. 1 | Niche adaptation is not apparent via total encoded gene architecture.** **a** Core gene phylogeny of 493 *S. pseudintermedius* isolates. Cohort, MLST, household (if applicable), and ARG burden are indicated by concentric color strips. **b**, **c** Jaccard dissimilarity by accessory gene content represented by **b** principal coordinate analysis ordination and **c** beta diversity compositions of within-group and between-group Jaccard distance. White, light gray, and dark gray violins indicate within cohort and niche, niche-only, and across cohort and niche pairwise comparisons, respectively, between 94 animal diagnostic, 163 human diagnostic, 90 pet colonizing, 33 human colonizing, and 113 environment isolates. Box plots extend from the 25th to 75th percentile with median line displayed, and whiskers extend to 1.5 x inter-quartile range. **d** *S. pseudintermedius* pangenome composition. **e** Total gene content breakdown by COG category.

smaller European studies[22,37], and a lack of representation of such diversity within our cohort in the PubMLST database. Together we observe 315 unique allelic combinations among our *S. pseudintermedius* isolates, 237 of which are previously uncharacterized. We have consequently added ST2385-ST2621 to the PubMLST database, expanding the total number of reported sequence types by 9.9%. Within our cohort, the most well-represented sequence types include previously known ST759, ST181, and ST923 (*n* = 9, 6, and 6 isolates, respectively), and newly added MSSP sequence types ST2553, ST2550, ST2558, ST2414, ST2606, ST2544, and ST2545 (*n* = 11, 9, 9, 7, 7, 6, and 6 isolates, respectively). As expected, most of these STs exclusively represent isolates from the household arm of our cohort, reflecting the multiple sites of collection and multiple collection timepoints for each household. Exceptions include ST923 and ST2414, which represent a mix of human diagnostic, animal diagnostic, and household isolates, and ST181, which exclusively represents human diagnostic isolates. ST181 is of unique clinical relevance as a known MRSP lineage that has previously been reported in colonizing and diagnostic human and canine isolates from Israel and Thailand, as well as Alberta, CA, and New England, USA; this is the first report of its identification in the American Midwest[13,21,38,39]. Notably, we also report the first observations of MRSP within ST862 (*n* = 5 isolates), all of which were captured in the household cohort. These data represent early indications of the uniqueness of our captured isolates relative to published *S. pseudintermedius* genomes.

To determine the full population structure of *S. pseudintermedius* and assess for phylogenetic separation based on niche or species-host, we annotated all protein coding sequences of each isolate assembly using Prokka and PGAP, and generated a species-specific core gene alignment with Roary. From this, we generated a maximum-likelihood phylogenetic tree reflecting 1704 genes present in ≥99% of isolates at ≥95% identity. We did not observe clear clustering of isolates by niche-type (Fig. 1a) or host-species (Supplementary Fig. 3A), though we do note two clades with high ARG burden that appear predominantly diagnostic in origin. We similarly constructed a core genome alignment to further consider common intergenic regions (Supplementary Fig. 4), with the resulting phylogeny further confirming no common core genome signature among isolates by niche-type or host-species. Isolates do appear to cluster locally by MLST and by household, with the most common sequence types often representing multiple isolates from the same household, though a larger relationship between pairwise ANI and geographic distance did not materialize ($R^2 = 0.019$, Supplementary Fig. 5).

Upon expanding our analysis to accessory gene content present in 1–99% of isolates, we again did not observe any clustering by niche-type or host-species (Fig. 1b, Supplementary Fig. 3B). An in-depth examination of Jaccard distances between the accessory gene content of isolates of the same cohort and niche, niche-only, and across cohorts and niches further confirmed a lack of differentiation by niche-type (Fig. 1c); the same was observed when compared by host-species (Supplementary Fig. 3C). This is likely due to the magnitude of cloud genes (those

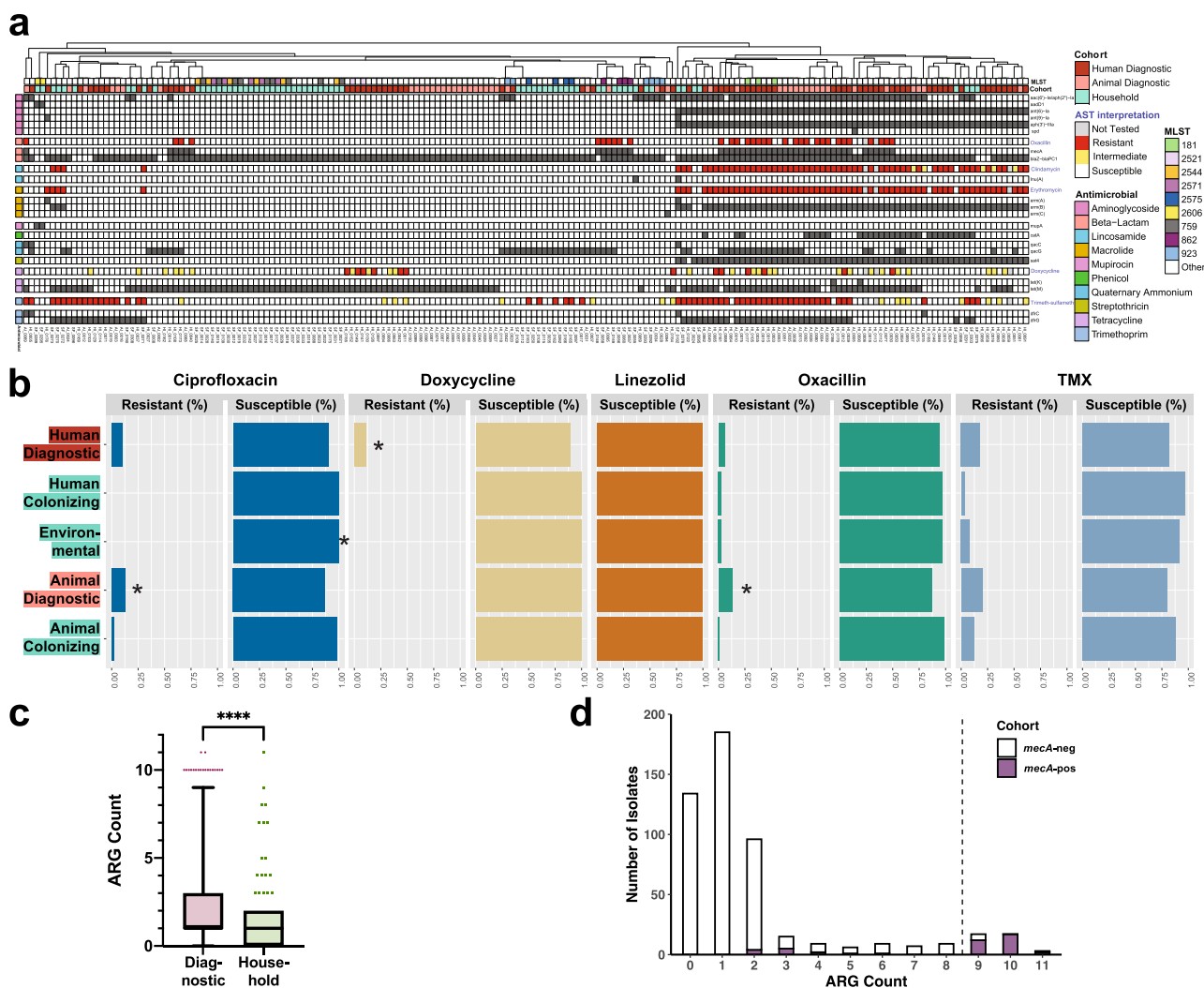

**Fig. 2 | Diagnostic isolates are distinguished by greater ARG burden and phenotypic resistance. a** Antibiotic resistance gene content and antibiotic susceptibility testing results, organized by antibiotic class. Only isolates with ≥2 ARGs (*n* = 188) are shown. **b** AST interpretation for commonly-prescribed antibiotics (Abbreviation TMX, trimethoprim-sulfamethoxazole). Asterisks indicate a cohort is significantly more resistant or susceptible to the respective antibiotic relative to all other isolates (*q* < 0.05, Chi-square). **c** ARG count for diagnostic and household

isolates (*n* = 255 and 238 isolates, respectively; *p* = 3.452e-09, two-sided Mann–Whitney). Box plot extends from the 25th to 75th percentile with median line displayed, and whiskers represent 10–90 percentiles. **d** Presence of *mecA* among isolates (*n* = 493), distributed by total ARG count. *mec*-positive isolates are significantly overrepresented among high ARG burden isolates (≥9 ARGs) relative to low ARG burden isolates (2–8 ARGs).

present in ≤15% of isolates) within the *S. pseudintermedius* pangenome, implying a typical genome to be comprised of the 1673 core genes, and a few hundred of the ~5000 cloud genes that are unique to each isolate or a handful of isolates (Fig. 1d). Finally, upon categorizing all encoded genes within each *S. pseudintermedius* assembly into functional COG (Cluster of Orthologous Genes) classifications, we again observe a nearly indistinguishable breakdown of gene content across cohorts (Fig. 1e). Collectively, we do not observe any niche differentiation among *S. pseudintermedius* by global encoded gene content.

## Two clades of diagnostic isolates are distinguished by greater ARG burden and phenotypic resistance

We observed that two core gene clades of diagnostic isolates, including MRSP ST181, are associated with a substantial ARG burden (Fig. 1a, green color strip). Comparing across niches, we find that diagnostic isolates have significantly more ARGs than isolates of household origin (*p* = 3.452e−09, two-sided Mann–Whitney; Fig. 2a, c). Many diagnostic isolates, regardless of host-species, harbored antibiotic resistance

genes (ARGs) in several antimicrobial classes (Fig. 2a, Supplementary Data 3). This ARG overabundance among diagnostic isolates translates to greater rates of phenotypic resistance: human diagnostic isolates are significantly more resistant to doxycycline (*q* < 0.05, FDR-corrected Chi-square) and often more resistant to ciprofloxacin (*q* = 0.075, FDR-corrected Chi-square) relative to all other isolates, animal diagnostic isolates are significantly more resistant to ciprofloxacin and oxacillin (*q* < 0.05, FDR-corrected Chi-square), and isolates captured on household surfaces ("Environmental") are significantly more susceptible to ciprofloxacin relative to all other isolates (*q* < 0.05, FDR-corrected Chi-square; Fig. 2b). Notably, isolates with greatest ARG burden (≥9 ARGs) were significantly more likely to also carry *mecA* compared to those of lesser burden (2–8) (*p* < 0.0001, Chi-square; Fig. 2d), in line with a report on 38 canines finding elevated ARG burdens among MRSP[25]. A minority of these phylogenetically-diverse isolates carrying the *mecA* gene presented as oxacillin-susceptible MRSP (OS-MRSP), likely due to the absence of encoded *mecA* regulators *mecI* and *mecR1* (Fig. 2a).

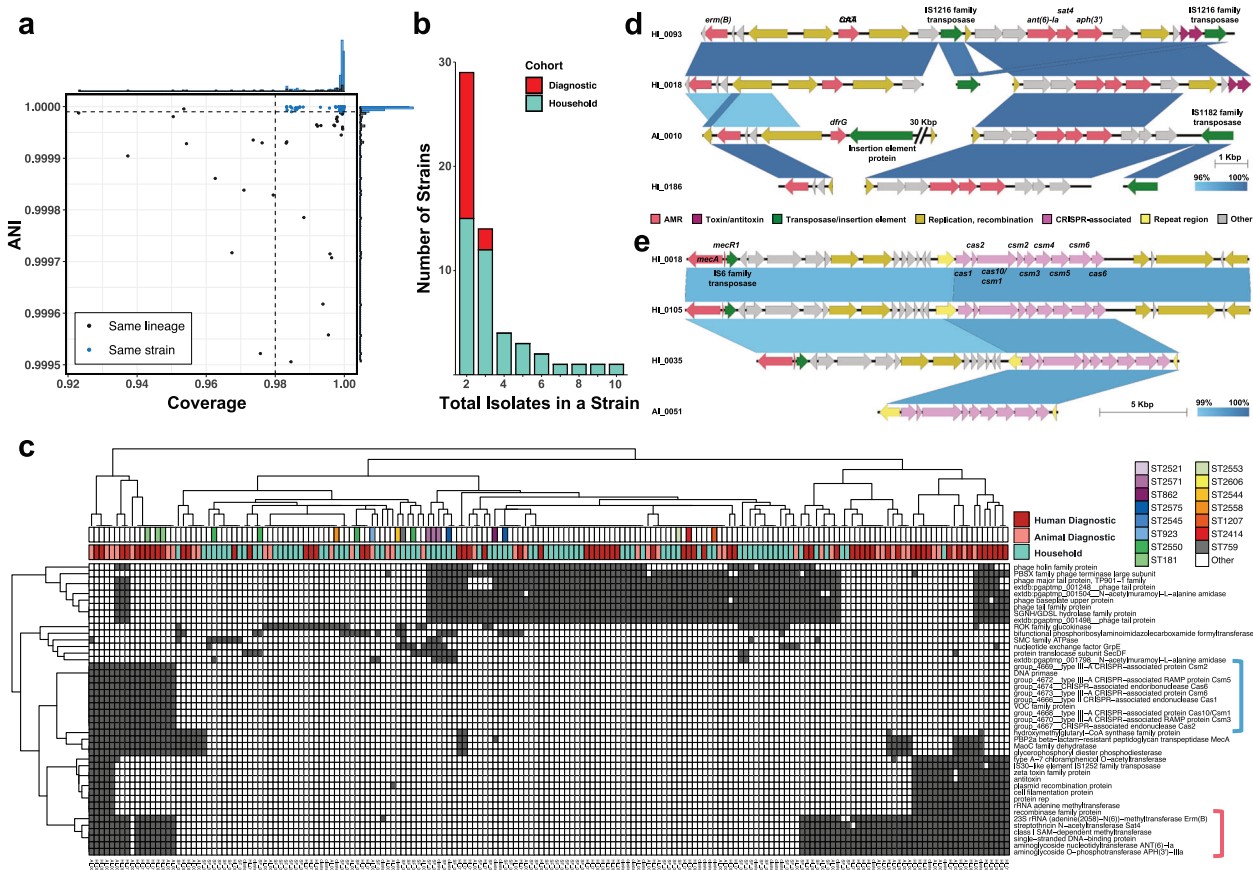

**Fig. 3 | Syntenic gene structures continue to distinguish select diagnostic isolates following strain correction. a** Pairwise comparisons of ANI and coverage within lineage clusters. Most comparisons reside in the top-right quadrant (>99.999% ANI, >98% coverage), indicating most isolates of the same lineage are members of the same strain cluster. **b** Distribution of strain clusters, by number of isolates within each cluster and cohort of origin (*n* = 56 clusters comprising 184 isolates). **c** Genes found by GWAS to be overrepresented among diagnostic or household isolates and strain cluster representatives. Only isolates or strain clusters with at least one GWAS-identified, overrepresented gene are displayed. Blue and red brackets denote a CRISPR-Cas system and resistance gene cluster, respectively. **d, e** Representative appearances of the **d** *Tn*5405-mediated resistance gene cluster and **e** type-IIIa CRISPR-Cas system among diagnostic isolates.

## Syntenic sets of cellular defense genes distinguish specific clusters of diagnostic *S. pseudintermedius* isolates

In consideration of the repeat sampling of households, and the temporal proximity between some human and animal participants in the clinic, we assessed for persistence of *S. pseudintermedius* strains in and across community and diagnostic settings. For this, we employed snp-sites and snippy to identify strain clusters defined as isolates (1) reporting >99.999% ANI (≤25 single nucleotide polymorphisms [SNPs] across all shared nucleotide positions) and (2) sharing ≥98% assembly coverage. Isolates that did not meet these criteria but were within 600 core gene SNPs of another were considered to be of the same lineage (Supplementary Fig. 6A, Fig. 3a). These efforts revealed widespread interrelatedness, capturing 230 of the 493 *S. pseudintermedius* isolates (*n* = 186 within strain clusters and *n* = 44 at lineage classification) (Fig. 3b, Supplementary Fig. 6B). The largest strain clusters (comprising 6–12 isolates per cluster) were composed entirely of isolates from the household cohort. Of significance, however, approximately half of the smallest strain clusters (capturing 2–3 isolates per cluster) comprised exclusively of human diagnostic isolates, all but one of which are MSSP. Within these strain clusters, isolates from different patients are captured 2–212 days apart (median: 25 days), implying some patients presenting at Barnes-Jewish Hospital are members of the same diagnostic *S. pseudintermedius* transmission networks. While our stringent strain cutoffs precluded appearances of any animal diagnostic isolates within strain clusters, or the observance of mixed human diagnostic-household strain clusters, expanding our analysis to

lineage clusters found the appearance of both (*n* = 7 animal diagnostic isolates, each a member of a different mixed cluster; *n* = 10 mixed clusters total; Supplementary Fig. 6B). Appearances of lineage clusters encompassing mixed household and diagnostic isolates (2–5 isolates per cluster) further extends the transmission potential noted in human diagnostic-exclusive strain clusters and signifies that the two niches are not mutually exclusionary, suggesting that crossover events involving a recent common ancestor may have seeded isolates of the same lineage in colonizing and diagnostic settings.

Capturing isolate interrelatedness also enabled a phylogenetically-informed genome-wide association study (GWAS) between household and diagnostic isolates, in which isolates of the same strain cluster were first collapsed into and represented by a single cluster-wide pangenome to avoid artificial signal from strain overrepresentation. These efforts identified two groups of genes that were entirely or nearly entirely absent in the genomes of household isolates (Fig. 3c). First, we observed exclusively in diagnostic isolates the known *Tn*-5405-mediated resistance gene cluster (ant(6)-Ia, sat4, aph(3')-IIIa)[21,26,40–45] encoding resistance to aminoglycosides and streptothricin, flanked by either IS1182- or IS1216- family transposases (Fig. 3d). The resistance gene cluster was often found alongside a toxin-antitoxin system and was occasionally positioned directly downstream of *erm(B)* and CAT, genes that traditionally encode resistance to macrolides, lincosamides, and streptogramin B[46], and chloramphenicol, respectively. Notably, when not in proximity to the resistance gene cluster, *erm(B)* was often found alongside trimethoprim-resistance gene *dfrG*. We also observed

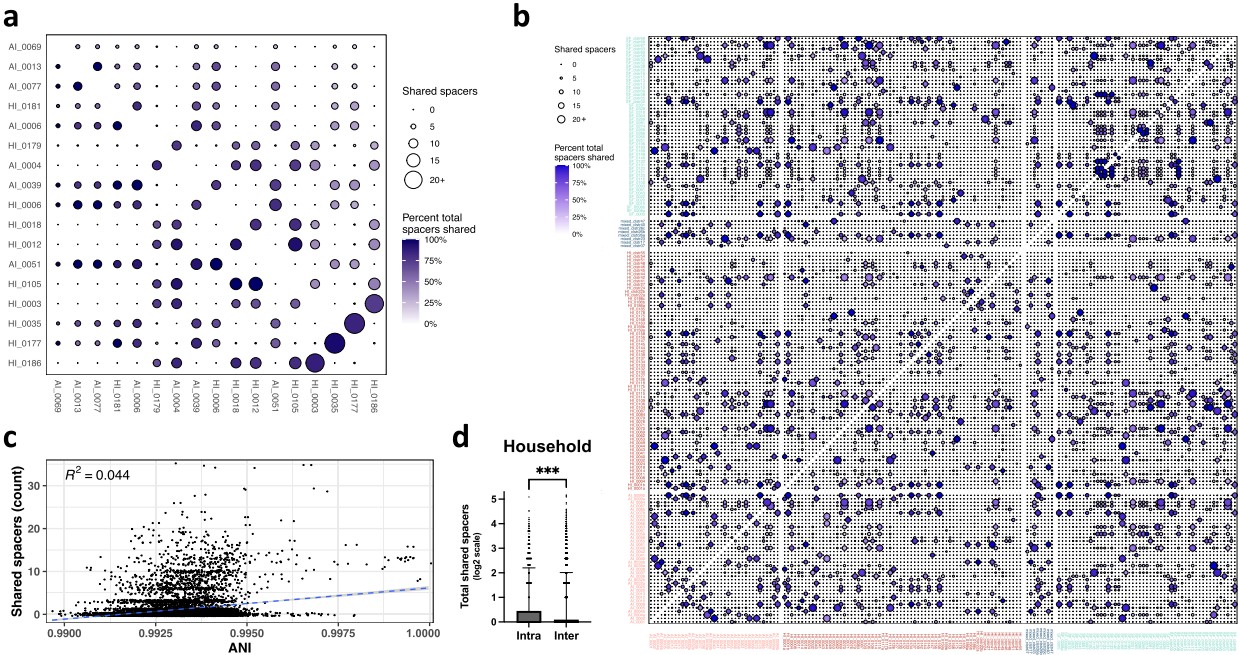

**Fig. 4 | Household isolates share more spacers with isolates within their niche than with diagnostic isolates, agnostic of pairwise ANI. a** Co-occurrence matrix of spacers shared at 100% ANI between only the 17 *mecA*-positive isolates with Type-IIIa CRISPR systems. Bubble size and saturation reflect total quantity of spacers and percent of spacer repertoire shared between two isolates, respectively. HI_0003-HI_0186 and HI_0035-HI_0177 are two examples of isolate pairs that fall into lineage but not strain clusters, yet have virtually identical spacer repertoires. **b** Co-

occurrence matrix of spacers shared at 100% ANI between all CRISPR-positive isolates. Isolate names and intra-cohort pairwise comparison bubbles are colored by cohort. **c** Pairwise ANI by shared spacer count. **d** Pairwise shared spacer counts of household isolates with other household isolates (intra), and with diagnostic isolates (inter) (*n* = 52 and 104 household and diagnostic isolate and cluster representatives, respectively; *q* = 0.00120, FDR-corrected Mann–Whitney). Barplot and error bars represent mean + SD.

a variant of the type-IIIa CRISPR system containing *cas* genes for Cas1-2, Csm2–6, and Cas10/Csm1 in 17 phylogenetically-diverse diagnostic isolates (Fig. 3e). Of significance, all 17 isolates are also *mecA*-positive, and in at least three isolates we observe the CRISPR system to be adjacent to the SCC*mec* element. While this type-IIIa system is found in an additional 48 isolates across the three cohorts, proteins in the systems of these 17 isolates cluster at ≥95% amino acid identity (AAI) with each other and ≤78% AAI with those in the other 48 isolates. Taken together, we observe a subset of diagnostic isolates of different lineages that are distinguishable from household *S. pseudintermedius* through their accumulation of genes involved in cellular defense against antibiotics and phages.

### Household isolates are more likely to share CRISPR spacers with other household isolates, agnostic of overall ANI

In addition to the two variants of the type-IIIa CRISPR system, a type II CRISPR system harboring the *cas9-cas1-cas2* genes is more commonly found among isolates across cohorts. Altogether, CRISPR-positive isolates (*n* = 238) harbored an average of 1.1 +/− 0.3 CRISPR systems, as some isolates harbored both type II and IIIa systems, and 15.4 +/− 5.6 spacers per CRISPR system (range 4–36 spacers) as identified by CRISPRCasFinder[47]. These CRISPR spacers—segments of nucleic acid from past foreign invaders incorporated into bacterial genomes—convey an immunologic memory of prior bacteriophage infections and presently integrated prophages, and form the basis of an adaptive immune response engendered by the linked Cas proteins[48]. Such prevalence of CRISPR systems and spacer content offers a window into the evolutionary histories of phage predation for each isolate, enabling a comparative analysis on shared or diverging exposures to natural predators within and across cohorts.

Towards this, we generated a co-occurrence matrix for all unique spacers shared between CRISPR-positive isolates at 100% ANI

(Fig. 4b). In consideration of phylogeny, all isolates of the same strain cluster were again condensed into a single cluster-representative pan-spacer collection prior to the spacer content comparisons. However, we noticed that in all cases, isolates that fell only into lineage clusters also shared all or nearly all of the spacers as their corresponding strain cluster-representative (Fig. 4a), so these representatives were expanded to lineage cluster-representatives for downstream analyses. We found high-identity spacer sharing to be infrequent between lineage clusters or unassigned isolates, with most pairwise comparisons sharing zero spacers between them. Nonetheless, 27.9% of pairwise comparisons shared at least one spacer; for these, an average of 3.7 spacers were shared, with up to 36 spacers shared between phylogenetically diverse isolates or lineage cluster representatives. Despite accounting for lineage associations, we initially speculated that spacer sharing was a function of isolate relatedness, with phylogenetically similar isolates expected to share a more recent common ancestor and therefore a longer timeline of shared evolutionary and phage predatory histories; surprisingly, this did not appear to be the case, as there was only a very weak correlation ($R^2$ = 0.044) between ANI and shared spacer count (Fig. 4c). However, when comparisons were binned based on cohort identity, we observed that household isolates were significantly more likely to share spacers with other household isolates (*q* = 0.00120, FDR-corrected Mann–Whitney) than human or animal diagnostic isolates (Fig. 4d). Environmental exposure, rather than phylogenetic similarity, dictated spacer sharing, indicating niche separation is present among household *S. pseudintermedius* in the context of host defense.

### Persisting strains within households evolve defense genes amidst MRSA decolonization interventions

Core gene analysis revealed that some household isolates cluster together phylogenetically and constitute reoccurring strains across

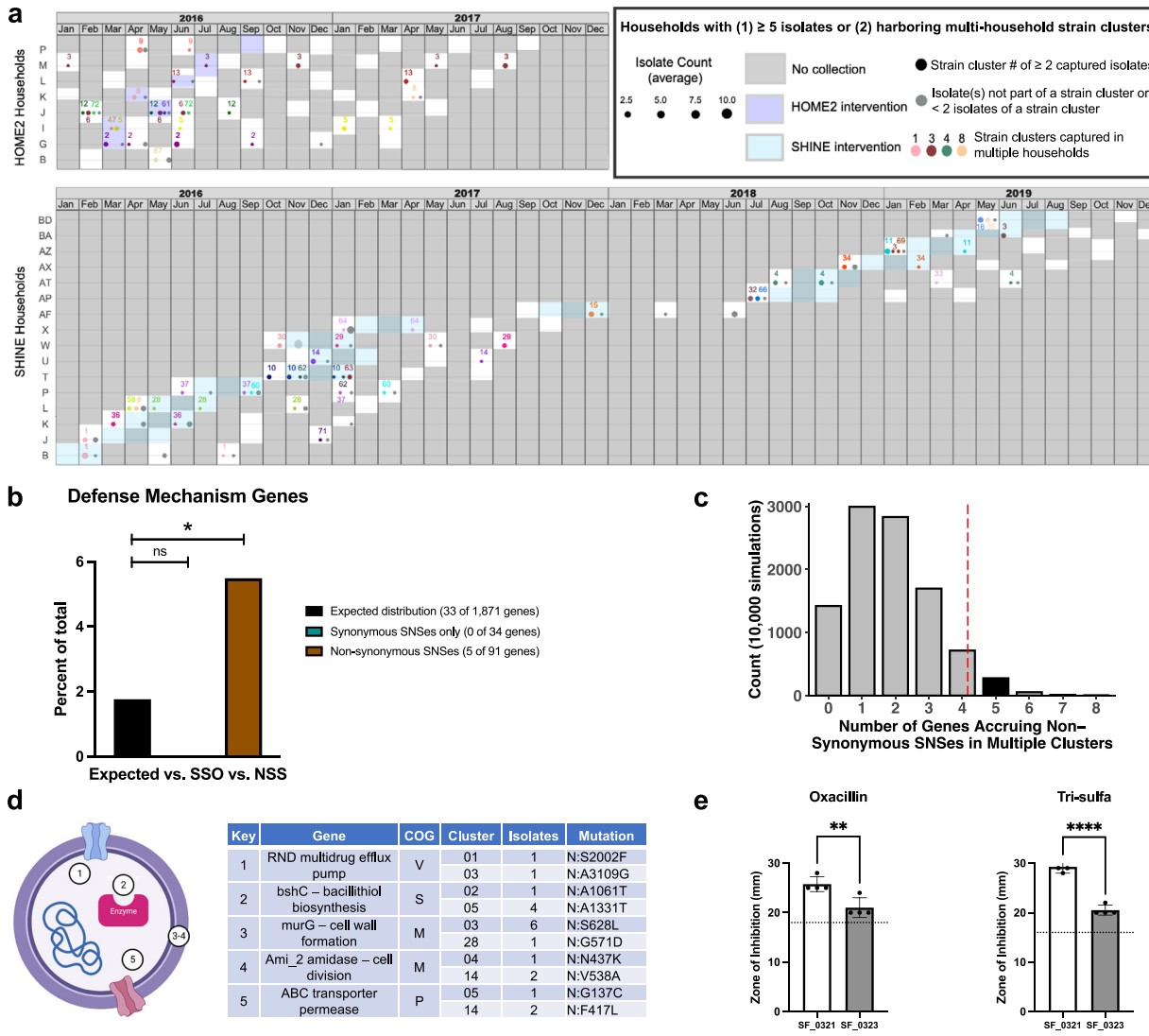

**Fig. 5 | Persisting strains accrue mutations in defense mechanism genes amidst household MRSA decolonization. a** Observance of strain clusters across households, faceted by subcohort. MRSA decolonization is highlighted and months without sample collections are noted in gray. Samples collected during the highlighted HOME2 intervention were collected immediately prior to MRSA decolonization and are considered baseline samples. Samples collected during the highlighted SHINE intervention were collected during MRSA decolonization. Bubble size corresponds to the number of isolates per strain cluster found in a collection month. **b** Defense mechanism genes as a percent of total genes accruing synonymous SNSes only (SSO) and non-synonymous SNSes (NSS), compared against the expected distribution of a reference *S. pseudintermedius* genome ($q = 0.0456$, FDR-adjusted one-sided Binomial test). **c** Number of genes accruing non-synonymous SNSes in multiple clusters, over 10,000 iterations. 9500 iterations are captured to the left of the dashed vertical line. **d** Schematic representation of the five NSS genes mutated in multiple strain clusters. COG key: [M] Cell wall/membrane/envelope biogenesis, [P] Inorganic ion transport & metabolism, [S] Function unknown, [V] Defense mechanisms. **e** AST results for oxacillin and trimethoprim-sulfamethoxazole ($q = 0.009$ and 9.878e−05, respectively, FDR-corrected two-sided unpaired T-test) for Cluster 03 isolates before and during household MRSA decolonization.

longitudinal samplings (Fig. 1a). Such households resemble microenvironments within which persistence, replacement, and/or reappearance of strain clusters are observable. Of the 44 households in the HOME2 and SHINE sub-cohorts, 24 (54.5%) harbored ≥5 *S. pseudintermedius* isolates and/or strain clusters found across multiple households (Fig. 5a). Strain clusters were typically household-specific, though several were isolated in multiple households and are highlighted (Fig. 5a, strain clusters 01, 03, 04, 08). These data reveal extensive *S. pseudintermedius* diversity in the community and the ability of strains to spread among inhabitants and surfaces once found within a household.

Each household underwent a period of attempted decolonization, in which mupirocin, chlorhexidine/dilute bleach water, and/or Clorox were applied to human inhabitants or household surfaces[34,35]. Though intended to interrupt community-acquired MRSA transmission

networks, these interventions present a real-world opportunity to observe the presence and extent of putative adaptive evolution within *S. pseudintermedius*. We hypothesized that such decolonization efforts would result in evolved genomic responses in the form of an accrual of non-synonymous−base changing−single nucleotide substitutions (SNSes) in staphylococcal defense genes.

To this end, we leveraged high depth of sequencing to track SNS accumulation in ten household strain clusters that persisted through the decolonization period, together capturing 45 isolates. These strain clusters comprise an average of 4.5 +/− 2.2 isolates collected over 6.8 +/− 3.3 months across 1.3 +/− 0.5 households (range 2–8 isolates, 3–12.4 months, and 1–2 households per persisting strain cluster). For each cluster, a high-quality isolate assembly from the initial month of detection was chosen as the cluster reference, against which reads from all subsequent isolates within the strain cluster were mapped. We

then ran inStrain to identify genes in subsequent isolates that accrued at least one SNS, together capturing 125 function-annotated genes across 270 reference-subsequent isolate comparisons (the same gene will appear in multiple comparisons if the mutation appeared early in the strain cluster and was maintained over time). A total of 91 genes harbored mutations that resulted in at least one non-synonymous SNS (NSS), while the remaining 34 genes carried mutations resulting in synonymous SNSes only (SSO) (Supplementary Data 5). These categories represent adaptive and neutralizing/purifying selection, respectively[49].

A total of 5 of the 91 NSS genes and 0 of the 34 SSO genes were annotated by COG as defense mechanisms. Compared against the distribution of defense mechanism genes within a representative *S. pseudintermedius* genome, we observed a significant over-representation of defense mechanism genes within the NSS dataset ($q = 0.0456$, FDR-adjusted one-sided Binomial test) (Fig. 5b). This relationship was unique to genes undergoing adaptive evolution, as the lack of SSO defense mechanism genes reflected the expected distribution ($q = 1$, FDR-adjusted two-sided Binomial test). Importantly, there was no difference between the overall COG distribution of genes accruing SNSes ($n = 125$) and the expected COG distribution of the full reference assembly ($p = 1$, Multinomial goodness of fit), nor was there an overall difference between the COG distribution of NSS and SSO genes ($p = 0.59$, Fisher's Exact test) (Supplementary Fig. 7). Taken together, the collection of genes that have accrued mutations resulting in non-synonymous substitutions over time and through decolonization do not appear to collectively be different in annotated function from the genes collecting synonymous substitutions only, and both gene populations together resemble the overall distribution of functional annotation of genes seen in a complete *S. pseudintermedius* assembly. The NSS genes are distinguished from the expected distribution solely by their elevated presence of defense mechanism genes accruing base-changing mutations.

Of note, we observed that five of the 101 NSS genes were mutated in more than one strain cluster. A permutation test found that this gene count to be significantly outside the expected count, indicating the five genes collecting non-synonymous mutations to be chosen non-randomly (Fig. 5c). These genes include a multidrug efflux pump, *bshC*, *murG*, *ami_2*, and an ABC transporter permease (Fig. 5d). Interestingly, while the efflux pump was the only gene to be formally annotated as a defense mechanism, *bshC* has a role in antifolate defense[50], *murG* catalyzes cell wall synthesis[51,52], and *ami_2* is involved in endolysis[53]. Most notably, within strain cluster 03 we observed a mutation in *murG* to appear alongside a significant reduction in susceptibility to oxacillin and trimethoprim-sulfamethoxazole ($q = 0.009$ and 9.878e−05, respectively, FDR-corrected two-sided unpaired T-test) (Fig. 5e).

## Discussion

We present a strain-resolved comparative genomics investigation of 493 *S. pseudintermedius* isolates captured across the American Midwest. These isolates were collected across multiple host-species and exist both commensally on human and pet skin and nares, pet fur, and on household fomites, as well as pathogenically within human and animal wounds, SSTIs, and respiratory and urinary tract infections. Contrary to what has been observed with *S. aureus*[54], we do not observe *S. pseudintermedius* clustering by core gene similarity or accessory gene content along host-species or niche delineations, or by geographic distance. This lack of correlation between pairwise ANI and geographic distance, coupled with our identification of over 300 unique multilocus sequence types, emphasizes an exceptionally diverse genomic landscape of *S. pseudintermedius*. Together, our findings suggest that the spread of *S. pseudintermedius* across multiple hosts and environments does not reflect longstanding lineage-based acclimation to host-species or niche.

Human and animal diagnostic isolates did, however, report significantly greater ARG carriage relative to those found in household settings, the first genomic indication of niche differentiation in this study. This is in part due to a transposon-mediated resistance gene cluster identified by strain-corrected GWAS to be overrepresented among diagnostic isolates. Comprehensive testing of all isolates against 12 antibiotics similarly revealed significantly greater levels of phenotypic resistance against ciprofloxacin, doxycycline, and/or oxacillin among human and animal diagnostic isolates. These antibiotics or antibiotic classes are first- and second-line therapeutics empirically prescribed for most clinical presentations that contextualize the human and animal diagnostic *S. pseudintermedius* cohorts[55], exemplifying an alarming trend that the *S. pseudintermedius* isolates found in clinical contexts are also the most resistant to the antibiotics frequently prescribed to resolve them[56]. Isolates with the greatest ARG burden were also significantly more likely to be MRSP, further highlighting their clinical importance. Importantly, a minority of *mecA*-positive isolates presented as OS-MRSP likely due to the absence of transcriptional regulator *mecR1*, *mecI*, or both[57].

A total of the 238 of the 493 (48.3%) *S. pseudintermedius* isolates across our cohorts harbored at least one CRISPR system, far greater than the 8–20% carriage rates observed by others[41,58]. We investigated sharing of CRISPR spacers between isolates, given their role as immunologic memory reflecting history of phage predation. Spacer-sharing was infrequent among isolates, an expected result given the range of bacteriophage invaders and the individualized trajectory of phage exposure and incorporation into immunologic memory. Still, 27.9% of pairwise comparisons shared at least one spacer. Remarkably, of these we observed that environmental exposure, rather than genomic relatedness, impacted the magnitude of shared spacers between pairwise comparisons. This trend was especially apparent among household isolates, which shared significantly more spacers with each other than with diagnostic isolates, reflecting for the first time a form of niche adaptation among commensal household *S. pseudintermedius*. Interestingly, like the ARG enrichment seen in diagnostic isolates, this niche response also appeared in the context of microbial host defense.

The household cohorts were originally designed to study decolonization protocols against community-associated MRSA, but also potentiated off-target collateral selection against coexisting *S. pseudintermedius* populations. Through our longitudinal sampling and sequencing, we were able to observe the genomic correlates of such interventions in real-world settings, which materialized through accrual of non-synonymous SNSes in genes specifically annotated as microbial host defense. We further observed five genes accruing these non-synonymous SNSes in multiple strain clusters in different households, four of which have direct or indirect roles in host defense. Ultimately, we observed during one decolonization intervention that strain cluster 03, which collected a non-synonymous SNS in cell wall formation gene *murG*, experienced a significant drop in susceptibility against oxacillin and trimethoprim-sulfamethoxazole. These results suggest that decolonization interventions may reduce drug susceptibility of some persisting *S. pseudintermedius*, making them more phenotypically akin to the diagnostic isolates collected in the hospital and veterinary clinic.

Our work is not without limitations. All isolates were captured in the American Midwest; though we observe considerable diversity by MLST and core gene architecture, this has not been contextualized against a global background. Most of the companion animal isolates were from dog hosts but a few were obtained from cats. While we grouped these hosts into a combined companion animal category, animal species represent unique ecological niches that could shape the genetic population structure of *S. pseudintermedius*, and warrant further investigation. Repeat sampling of households within and across timepoints increases the risk of clonal overrepresentation when comparing against single isolate representatives from human and

animal wounds and infections. While we have performed considerable mitigation efforts via generation of strain- and lineage-cluster representatives in our GWAS and CRISPR spacer screen, it is possible we were not completely successful. Conversely, not every *S. pseudintermedius* isolate captured in the household surveillance cohorts was sequenced, obfuscating our ability to differentiate strain persistence from disappearance and reintroduction within households. Lastly, while *murG* accrued a non-synonymous SNS in strain clusters 03 and 28, the isolate pair displaying a reduction in susceptibility in cluster 03 also accrued non-synonymous SNSes in several other genes, each of which warrant further investigation.

While others have conducted comparative genomics studies of *S. pseudintermedius* populations, all have so far been limited to one host-species and/or niche type[20–22,25,26,31,59,60]. Our study is the first to integrate whole-genome sequencing of *S. pseudintermedius* isolated from both human and animal hosts as a commensal and as an opportunistic pathogen, as well as isolates captured on abiotic surfaces within household environments. Together we have compiled the largest multi-center genomics-based cohort study of *S. pseudintermedius* to date and have added 237 new multilocus sequence types and 501 new SIG genome assemblies to publicly available databases (BioProject PRJNA908872). Within the context of existing literature, our work supports data by others that the *S. pseudintermedius* pangenome is overwhelmingly composed of shell genes present in <15% of isolates[20,21], likely influencing the lack of accessory genome signature by host-species or niche type. We extend reports of *S. pseudintermedius* captured on dogs with and without pyoderma[25] that isolates collected in diagnostic conditions carry a significantly larger ARG burden than those found commensally, with our work showing this to be agnostic of host-species. Others have previously been unable to identify host-specific genes distinguishing human and canine clinical isolates[22]; indeed, our GWAS indicates that *S. pseudintermedius* are distinguishable not by host species but instead by niche-type. There have been numerous reports of the ant(6)-Ia, sat4, aph(3')-IIIa resistance gene cluster and/or type-IIIa CRISPR system in some *S. pseudintermedius* isolate genomes[20–22,26,40–45,58,59], but our work is the first to show that these gene clusters are overrepresented among diagnostic relative to commensal isolates. With the inclusion of the type-IIc CRISPR system, our study is also the first to use spacer content to observe niche adaptation via putatively shared evolutionary exposures. Lastly, beyond *S. pseudintermedius*, we validate findings by others that MALDI-TOF MS has sufficient discriminatory ability to differentiate between SIG species[60], and add eight high-quality *S. delphini* genome assemblies to NCBI, a 23% expansion beyond existing resources. Altogether, this study is novel in that we have elucidated non-traditional niche differences within *S. pseudintermedius*, demonstrating parallel selection of defense mechanisms via gene acquisition in diagnostic isolates and gene mutation in household isolates, which would have not been uncovered by phylogenetic or phenotypic analyses alone.

## Methods

### Ethical approval
Washington University Institutional review board (IRB) and Institutional Animal Care and Use Committee (IACUC) approval was obtained for this study.

### Study cohort and identification
Human clinical specimens were cultured in the Barnes-Jewish Hospital Clinical Microbiology Laboratory in St. Louis, MO, USA, according to laboratory standard operating procedures. Isolates from these specimens (n = 181), primarily from human wounds, tissue, respiratory tract, and drainage, were collected as part of routine clinical care, with cultures submitted from patients with clinical symptoms suggestive of infection. Clinical specimens were plated to agar medium according to the laboratory's standard operating procedures for each specimen type. In general, specimens were plated to sheep's blood, chocolate, and MacConkey agar (Remel, Lenexa, KS, USA). Isolates of *S. pseudintermedius* were most commonly obtained from the sheep's blood agar plate. All human diagnostic isolate collection occurred between December 2011 and July 2019. The diagnostic canine and feline SIG isolates were selected from a collection of isolates recovered from submissions to the Kansas State Veterinary Diagnostic Laboratory. These isolates were recovered from specimens submitted for clinical culture, primarily from animal urinary tract and skin and soft tissue infections (n = 100), with primary isolation occurring between January 2016 and June 2019. Isolates recovered from nondescript specimen types ("swab"), duplicate submissions from the same veterinarian-owner combination (over the entire study period), and non-viable isolates were not eligible for inclusion in this study and were replaced with the next eligible isolate until a total of 25 isolates were identified per calendar year. Colonizing isolates (from human nares, axillae, or inguinal folds and from dogs nares, mouth, or dorsal fur) and household environmental surface isolates (e.g., bed linens, kitchen table, refrigerator door handle, bathroom countertop, bathroom faucet handles, bathroom light switch, toilet seat, bathtub, television remote control, computer keyboard and mouse, and telephone) were collected through two *S. aureus* surveillance projects among households with children with recent *S. aureus* infections – the Staph Household Intervention for Eradication (SHINE; NCT02572791) and Household Observation of MRSA in the Environment 2 (HOME2; NCT01814371) as previously described[34,35] (Pet = 108, Human = 31, Surface=151). All samples were selected from the first timepoint in which a SIG isolate was recovered from a specific individual, pet, or environment site. For human samples, the nares were prioritized, followed by the axillae, and then the inguinal folds. In pets, the nares or mouth sample was prioritized, followed by the dorsal fur sample. In each of these studies, participants performed a decolonization regimen to eradicate *S. aureus* carriage from the skin and nares, consisting of intranasal mupirocin application and daily chlorhexidine gluconate body washes or dilute bleach-water baths.

### Identification and resistance characterization
The identification of all isolates was confirmed using the VITEK MS MALDI-TOF MS system (bioMerieux, Durham, NC). Susceptibility testing against oxacillin, delafloxacin, ciprofloxacin, enrofloxacin, cefoxitin, doxycycline, linezolid, trimethoprim-sulfamethoxazole (TMP-SXT), rifampin, clindamycin, erythromycin, and eravacycline was performed for each isolate via disk diffusion on Mueller Hinton agar. Methods followed the procedural guidelines outlined by the Clinical and Laboratory Standards Institute (CLSI) in the M02 and interpreted using M100 (30th ed) and VET01 (3rd ed) standards[61,62,90]. Where CLSI breakpoints were not available, FDA breakpoints were used. Quality control was performed on each day of testing.

Pearson's Chi-squared test was used to measure differences in AST interpretations across cohorts for 513 isolates using the R STATS package. Total susceptible and resistant counts for each cohort were compared against the sum for all other cohorts. For each antibiotic, isolates were excluded if results were unavailable for the specific antibiotic (n = 30 for ciprofloxacin, n = 30 for TMX) or if an intermediate antibiotic interpretation was reported (n = 27 for doxycycline, only reported for human diagnostic samples). *P*-values were corrected using the Benjamini-Hochberg false discovery rate method with the *p.adj* function in the R STATS package (method = "fdr").

### Illumina whole-genome sequencing
Genomic DNA was isolated from cell-suspensions and sequenced as previously described[23,63]. Briefly, DNA was quantified with Quant-iT PicoGreen dsDNA assay (Thermo Fisher Scientific, Waltham, MA, USA), with 0.5 ng used as input for Illumina Nextera XT library preparation (Illumina, San Diego, CA, USA). Libraries were pooled at equal

concentrations and sequenced on the Illumina NextSeq 500 High-Output platform (Illumina, San Diego, CA, USA) to a depth of 1.5 million reads per sample (2 × 150 bp). FastQC was used to determine read quality[64]. Trimmomatic (version 0.38; flags: leading, 10; trailing, 10; sliding window, 4:15; and minimum length (minlen), 60) was used to remove adapter sequences and low-quality reads from demultiplexed data[65].

## Core gene and core genome analysis
Contigs were assembled using Unicycler (version 0.4.8) at default parameters[66]. Deep-sequenced assemblies were downsampled to 100x coverage using seqtk (version 1.3)[67]. Inclusion criteria of draft genomes for all downstream genomic analyses included >98% completeness, <2% contamination, 0% strain heterogeneity, N50 > 40 kbp, and >15x average coverage, as assessed by CheckM (version 1.0.13), QUAST (version 4.5), BBmap (BBtools version 1.0.1) and seqtk (version 1.3)[67-70].

Assembled contigs were annotated using the NCBI Prokaryotic Genome Annotation Pipeline (PGAP, version 5.3)[71,72]. Core and accessory gene content, and core gene alignment, for S. *pseudintermedius* and S. *delphini* was determined using Roary (version 3.12; flags: -g 500000, -e)[73]. A core genome alignment was similarly constructed using Parsnp (version 1.2; flags: -c, -r!)[74]. In silico multi-locus sequence typing (MLST) was performed for all isolates against the PubMLST database[75] using the mlst tool (version 2.19)[76]. 237 new multilocus sequence types representing 376 S. *pseudintermedius* isolates were uploaded and accepted to PubMLST under the submission ID BIGSdb_20220927223226_035094_56619 and BIGSdb_20230109221715_2606963_61712.

A maximum likelihood tree was constructed with FastTree (version 2.1.10) and visualized using iTOL[77,78]. Pairwise average nucleotide identity (ANI) values were obtained using fastANI (version 1.1)[36]. A total of 40 S. *pseudintermedius* assemblies designated as "Complete Genomes," along with all assemblies available at the time of analysis for S. *delphini* (n = 22), S. *intermedius* (n = 7), and S. *ursi* (n = 1) were downloaded from NCBI and used to determine species-level taxonomy of all isolates.

Zip codes of the residence of the hospital patient or household of each isolate were collected. Zip code metadata was unavailable for the animal diagnostic cohort, which was excluded from this analysis. Geographic distance between isolates was determined using the *geocode_zip* function (R zipcodeR package, version 0.3.4)[79]. This was plotted against pairwise ANI after removal of self- and repetitive-comparisons using the *ggscatter* function (R ggpubr package, version 0.4.0; flags: add = "reg.line", cor.method = "pearson", conf.int=TRUE, core.coef=TRUE) and ggmarginal (R ggExtra package version 0.10)[80,81].

## Accessory genome analysis
The gene_presence_absence.Rtab output file from Roary was purged of core genes (those present in >99% of isolates), resulting in 5910 accessory genes. Presence-absence of these were used to calculate Jaccard distance between all isolates using the *vegdist* function (R vegan package, version 2.5–7)[82]. Clustering by accessory gene content similarity was visualized through principal coordinate analysis using the *pcoa* and *ggplot* functions (R ape and ggplot2 packages)[83,84]. Jaccard dissimilarity scores were plotted following isolate metadata associations, including site of isolation and host-type. COG categories were assigned to all genes within each isolate assembly using eggnog (version 2.0.1; flags: -m diamond --query-cover 0.9)[85,86].

ARGs in the protein FASTA (.faa) output files from Prokka were annotated against the NCBI comprehensive database of acquired and intrinsic antimicrobial resistance proteins at >90% identity and >90% coverage using AMRfinder (version 3.9.8; flag: --organism Staphylococcus_pseudintermedius)[87]. A presence-absence heatmap was generated using the *pheatmap* function (R pheatmap; flag: clustering_method = 'mcquitty'), with columns clustered based on shared ARG content and rows organized by antimicrobial class[88].

Associated metadata, including cohort, AST interpretation, and antimicrobial class, are displayed as color strips. ARG count was compared between diagnostic and colonizing isolates (two-sided Mann–Whitney) and plotted in Prism 9, and *mecA*-positivity was tested for association with 'high ARG burden' status (Chi square) and plotted using the *ggplot* function.

## Strain cluster identification
Isolate-specific SNPs were called against the Roary-generated core gene alignment file using snp-sites (version 2.4.0)[89]. Isolate assemblies were then compared in an all-v-all manner based on their shared SNP content, with 600 core gene SNPs empirically determined to be the threshold for lineage-based relatedness (Supplementary Fig. 6A). Intra-lineage isolates were further compared using Snippy (version 4.4.3) for pairwise whole genome SNP calling[90]. Isolates were considered members of the same strain cluster if their whole genome SNP distance (Snippy VariantTotal) was below 26 (≥99.999% ANI) and overall coverage breadth ≥98%[91,92]. Accessory gene presence-absence data for isolates of the same strain cluster were collapsed into single strain-representative accessory genomes to control for phylogenetic over-representation in downstream analyses.

Phylogenetically-controlled genome-wide association analysis between diagnostic and household isolates was performed using Scoary (version 1.6.16) on the strain-informed gene_presence_absence.csv[93]. Presence-absence of genes with Benjamini-Hochberg-adjusted *p*-value < 0.05 were hierarchically clustered and visualized using the *pheatmap* function (R pheatmap and dendsort packages); isolates lacking all significant genes were removed from the heatmap, as were genes present in >100 isolates (n = 17, mostly tRNA alleles), or genes with nonspecific "domain-containing protein" annotations (n = 6)[88,94]. Gene structures significantly associated with a cohort were visualized using EasyFig (version 2.2.2)[95]. Towards this, GenBank files outputted by PGAP were first filtered to only the contig(s) carrying the genes of interest, and then preprocessed in ApE (version 3.1.2)[96].

## CRISPR spacer investigation
Contigs that harbored at least one PGAP-annotated Cas gene were extracted from each isolate assembly and added to an isolate-specific multifasta (one contig per isolate multifasta in 213/238 isolates), which was used as input for CrisprCasFinder[47]. Identified spacer sequences from each contig within each isolate were captured and labeled with their isolate and contig of origin and numeric position within the Cas operon. Reverse complements were generated using The Sequence Manipulation Suite[97]. All spacers and their reverse complements were inputted into clustalo (version 1.2.4; flags: --dealign --full --percent-id) to generate an all-v-all ANI-based spacer distance matrix[98]. Self-comparisons (spacers from a given Cas operon compared against themselves) were removed, and only the remaining spacer comparisons with ANI = 100% were retained for spacer-co-occurrence analysis. For the minority of isolates with Cas operons on multiple contigs, if this inter-operon, intra-isolate spacer comparison indicated an operon duplication, the second iteration was removed from analysis; if the two operons were unique, the title of each was amended with a suffix "a", "b", or "c" (i.e., HI_0186a, HI_0186b, HI_0186c) and both were retained. The subset dataframe was further reduced such that inverse entries were removed (i.e., if spacer 1 vs. spacer 2 is already present, spacer 2 vs. spacer 1 is removed). The cleaned dataframe was converted into an all-v-all isolate occurrence matrix plotting the number of spacers shared at 100% ANI between each isolate comparison, and visualized as a bubbleplot using the *ggplot* function. This was done at the isolate level and at the lineage level, after it was empirically observed that isolates of the same lineage had entirely or nearly entirely identical spacers. For the latter case, a lineage-representative displaying all spacers present in the underlying isolates was constructed and utilized.

The number of shared spacers between isolates was plotted as a function of ANI and visualized as a scatterplot with trendline (geom_smooth(method = "lm")) using the *ggplot* function. For each cohort type, isolate-isolate shared spacer counts were binned as intra-cohort or inter-cohort, compared (FDR corrected one-way Mann–Whitney) (wilcox.test(paired = FALSE, exact = FALSE, conf.int = TRUE, alternative = "greater"); R STATS package), and plotted in Prism 9.

### Within household tracking and evolution

To investigate strain dynamics within households, human colonizing, animal colonizing, and surface isolates were visualized within and between households and studies (HOME2 and SHINE). Strain clusters were plotted by household and faceted by study. Of the 44 longitudinally-sampled households, 23 met our primary inclusion criteria of harboring ≥5 isolates. One additional household harboring 4 isolates was also included as it harbored a strain cluster present in multiple households.

Isolates of the same strain cluster that appeared in households over a minimum time span of 3 months, were assembled at ≥30x coverage, and were visualized in Fig. 5a were evaluated for single nucleotide substitution (SNS) accumulation over time. For clusters that appeared in both SHINE and HOME2, only isolates from SHINE households were used to maintain consistency among decolonization exposures. If multiple isolates of the same strain cluster were captured in the index month, a reference isolate was chosen using the following formula: Completeness – (5 × Contamination) + ( Contamination × (Strain Heterogeneity/100)) + (0.5 × log(N50))[99]. Open reading frames (ORFs) within the reference assembly for each strain cluster were called via Prodigal (version 2.6.3)[100]. Reads for all subsequent longitudinally-captured isolates within a strain cluster were aligned to the strain cluster reference assembly using Bowtie2 (version 2.4.2)[101]. Each strain cluster's reference assembly and Prodigal-called ORFs, as well as the Bowtie2 alignment file for each subsequent isolate against the strain cluster reference, were inputted into inStrain-profile (version 1.5.7) to track SNS accumulation over time, as well as the genes they reside in[92]. The inStrain gene_info output file was reviewed to identify all instances of ORFs with at least one SNS (SNS_count ≥ 1; *n* = 182 unique genes). These Prodigal-called ORFs were united with their corresponding pgap annotations via blastp (BLAST+ version 2.12.0) and assigned a COG category via eggnog (version 2.1.8; flags: -m diamond --pident 90 --query_cover 80)[85,102]. Genes with blank, "-", or multi-letter COG categories, as well as genes with "partial" in their pgap annotation, were removed from further analysis, resulting in 125 remaining genes with SNS_count ≥1.

Genes were divided into two groups, depending on whether all harbored substitution(s) were synonymous (synonymous SNSes only; SSO), or if at least one substitution was non-synonymous (non-synonymous SNS(es); NSS). Observed COG distribution of NSS genes annotated as "defense mechanisms" was assessed for enrichment (one-sided Binomial test) (binom.test(alternative = "greater"); R STATS package). The distribution of COG categories among all genes accruing SNSes (SNS_count ≥1) was compared against the expected COG distribution of a full *S. pseudintermedius* assembly (Multinomial goodness of fit) (multinomial.test(useChisq = FALSE, MonteCarlo = TRUE); R STATS package). Overall distribution of COG categories was compared across the NSS and SSO groups (Fisher's Exact Test of Independence) (fisher.test(simulate.p.value); R STATS package).

A permutation test was implemented to determine whether the number of observed NSS genes mutated in multiple strain clusters was significantly greater than the expected count. Gene positions (numbered 1 to 1871, the total number of genes in the reference *S. pseudintermedius* assembly) were randomly sampled without replacement for each cluster, with the number of samplings equaling the number of genes that had undergone at least one non-synonymous substitution in that cluster (range 3–19 samplings, cluster-dependent). Samplings across clusters were combined and frequency of each gene position was determined. The total number of gene positions that were randomly sampled more than once were recorded. This process was iterated 10,000 times, and the frequency of gene positions randomly sampled more than once (0–8 total gene positions) was plotted in ascending order numerically. Gene counts that were greater than the value reported by the 9500th iteration was determined to be significantly outside the range expected from chance alone.

Two isolates in strain cluster 03 flanking decolonization reported drops in susceptibility against two antibiotics. Technical replicates of each isolate were retested against oxacillin and trimethoprim-sulfamethoxazole and results were compared using Student's T test. Results were plotted in Prism 9.

### Statistics and reproducibility

Unless otherwise stated, comparisons of categorical data were performed by Pearson's chi-squared test or Fisher's exact test depending on sample size while comparisons of continuous variables were tested using the Mann–Whitney U test. All tests were two-sided, and statistical significance was defined as $p < 0.05$. Multiple hypothesis correction was performed when appropriate using the FDR method. Multinomial goodness of fit was used to determine if the distribution of genes accruing SNSes by COG category differed from expected. Binomial test was used to determine if a specific COG category was enriched. A permutation test was employed to determine whether observed count of NSS genes appearing in multiple clusters was greater than expected. Statistical analyses were performed using the R STATS package or Prism 9.

### Reporting summary

Further information on research design is available in the Nature Portfolio Reporting Summary linked to this article.

## Data availability

All isolate sequencing data, including short reads and assemblies, have been deposited in the NCBI SRA and GenBank databases under BioProject PRJNA908872. Specimen metadata, antibiotic susceptibility testing data, assembly quality and genome metadata, and inStrain data are provided in Supplementary Data 1, 3–5, respectively. Certain raw metadata for human-origin isolates are protected and are not available due to data privacy laws; processed data presented in aggregate are available in Supplementary Data 2. Source data are provided with this paper.

## Code availability

All tools and R packages used for this analysis are publicly available and fully described in the "Methods" section. Detailed code is available at https://github.com/sanjsawhney/staphylococcus-pseudintermedius.

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

## Acknowledgements

This work was supported in part by awards from the Eunice Kennedy Shriver National Institute of Child Health and Human Development (NICHD; grant R01HD092414, G.D.) of the National Institutes of Health (NIH), the National Institute of Allergy and Infectious Diseases (NIAID; grant R01AI155893, G.D.; grant K23-AI091690, S.A.F.) of the NIH, the Agency for Healthcare Research and Quality (AHRQ; grant R01HS021736, S.A.F.; R01HS024269, S.A.F.) of the NIH, the National Center for Advancing Translational Sciences (NCATS; grant UL1-TR002345, S.A.F.) of the NIH, and the Children's Discovery Institute of Washington University and St. Louis Children's Hospital (S.A.F.). S.S.S. and R.C.V. are supported by the NIH-funded Training Programs in Cellular & Molecular Biology (grant T32GM007067, S.S.S.; PI Heather True-Krob; National Institute of General Medical Sciences) and Genome Analysis (grant T32HG000045, R.C.V.; PIs Michael Brent and Barak Cohen; National Human Genome Research Institute). The content is solely the responsibility of the authors and does not necessarily represent the official views of the funding agencies. We thank members of the Dantas laboratory, specifically Bejan Mahmud and Yao-Peng Xue, for helpful scientific discussions, and staff from the Edison Family Center for Genome Sciences & Systems Biology, including Eric Martin, Brian Koebbe, Jessica Hoisington-López, MariaLynn Crosby, and Bonnie Dee for technical and administrative support in high-throughput sequencing and computing.

## Author contributions

Conceptualization: C.-A.D.B. and G.D.; sample curation: B.V.L., S.A.F., and C.-A.D.B; MALDI-TOF MS, antibiotic susceptibility testing: M.A.W., C.E.M., and C.-A.D.B.; methodology, investigation, data curation, analyses: S.S.S. and R.C.V.; figure preparation and writing of manuscript: S.S.S. and R.C.V.; reviewing and editing of manuscript: All authors. Employment or leadership: C.-A.D.B., Journal of Clinical Microbiology, Clinical Microbiology Newsletter. Consultant or advisory role: C.-A.D.B., Bio Rad Laboratories, Thermo Fisher Scientific, Pattern, Beckman Coulter; G. Dantas, Viosera Therapeutics, Pluton Biosciences. Honoraria: C.-A.D.B., BioFire, Cepheid, Roche. Research funding: C.-A.D.B., bioMerieux, Cepheid, Luminex.

## Competing interests

G.D. holds US patent 10500191B2, Dec 10, 2019 (Composition and methods of use of antibacterial drug combinations), and declares stock ownership in Viosera Therapeutics. The remaining authors declare no competing interests.
