## [Peer Review File · Nature Communications]

REVIEWER COMMENTS

Reviewer #1 (Remarks to the Author):

Sawhney et al. present their genomic investigation of 493 *S. pseudintermedius* isolates sampled from animal and human clinical sources, colonized human and animals, and household fomites. The authors report gene acquisition (in diagnostic isolates) and mutation (in household isolates) act to differentiate isolates by niche type. Overall, this well-written manuscript presents interesting new findings regarding mechanisms (CRISPR systems and nonsynonymous mutations in defense genes) that contribute to niche differentiation and adaptation in co-circulating bacterial lineages. The methodology is sound and reproducible. The conclusions are supported by the data. My comments are listed below.

1. Font size in all figures was too small to be legible.
2. Although most pet isolates in the study came from dogs, a few were obtained from cats. Dogs and cats are also unique ecological niches that may also shape the population structure and differentiation of *S. pseudintermedius*. This must also be acknowledged and/or recognized as a limitation.
3. The rationale for using 25 whole genome SNPs (line 211) and 600 core genome SNPs (line 213) to define clusters or lineages need to be stated.
4. Avoid starting a sentence with numerical figure (e.g., lines 285, 310, 361). For example, instead of “5 of the 91 NSS genes...” (line 310), you can say “A total of 5 of the 91 NSS genes ...” or something similar.
5. Line 514: Missing close parenthesis
6. Intro, lines 65-66 “methicillin resistance – which is very closely tied to multidrug resistance”: Missing citation
7. Intro, line 112 “whole genome architecture does not track with host-species or niche”: The use of track in this sentence is confusing.
8. Supplementary table 1 should also include genomic features of each isolate, including N50, number of contigs, number of protein coding genes, % completeness, % contamination, genome length, date of isolate collection, body source or disease, accession number
9. Supplementary figure 4: legend should indicate what HI and SF represent.
10. It is unclear why *S. delphini*, *S. intermedius* and *S. ursi* were included in the analysis.
11. Methods, line 431: How the diagnostic canine and feline SIG isolates were selected from the larger submission collection must be stated. It was also no clear whether each isolate included in the study represented a single individual, or were there instances of multiple isolates from the same human or animal? In either case, this should be clarified.

Reviewer #2 (Remarks to the Author):

NCOMMS-23-05743

The authors sequenced 672 strains of *Staphylococcus* of the *intermedius* group (SIG) using Illumina. After quality filtering of the sequence reads, 501 were retained as *Staphylococcus pseudintermedius* for further analysis. The strains were of different human, animal and environment origins and were classified as household isolates and diagnostic isolates. The authors performed WGS-based phylogenetic and comparative analysis as well as antibiotic resistance gene detection and CRISPR identification to determine if differences exist between the isolates from the different origins and niches. Additionally, they analyzed a set of isolates which were isolated from households which were part of a *S. aureus* decolonization program. This study generated a large amount of data and gives new insights into the genomics and genetic diversity of *S. pseudintermedius*. Genomic analysis has been well performed. However, analysis of the resistance mechanisms and associated mobile genetic elements needs to be improved. Presence of virulence factors and prophages should be investigated to support the conclusion.

L39. Replace 572 with the actual number of *S. pseudintermedius* used for analysis (n=501)(see L129). Otherwise, it is confusing.

L45. CRISPR systems were also present in clinical isolates, mainly associated with SCCmec elements. Why only mentioning the household isolates here?

L46. Genes associated with resistance to oxacillin and trimethoprim. This cannot be said as long as it has not been proven experimentally. This is a speculation and should be deleted from the abstract.

L67. "methicillin-resistant"

L78. Define MSSP. The authors should also clearly differentiate their isolates as MSSP and MRSP in the study. These specifications would help to better distinguish between antibiotic-susceptible and -resistant isolates.

L111-L118. This part is a summary of the results and should be removed from the Introduction.

L116. What defines a defense gene? Against what?

L122. "staphylococci"

L122-L138. This part consists mainly of M&M and should be shortened. Only results should be presented.

L136-L138. This sentence is already mentioned in the M&M section. Avoid repetition.

L154. Delete "surprising". It is well known that the diversity is higher among MSSP.

L157-L165. This paragraph should better differentiate between the STs which have been found in MSSP and the STs of MRSP. It should also be mentioned if more diverse STs were found in MSSP than in MRSP.

Which STs are associated with MRSP. Are they known? Is ST181 the only ST associated with MRSP in this study?

L174. Delete surprisingly.

L174. Any link between clusters and ST?

L188. Define COG

L193. Here again: Are the isolates belonging to these clades MRSP? Do they belong to specific STs?

L200. Add fluoroquinolone phenotype and genotype in Figure 2. Do the veterinary diagnostic lab test for ciprofloxacin? Please check if the strains have mutations in the quinolone resistance-determining region of *gyrA* and *parC*.

L201. Based on Figure 2, the abundance of *mecA* gene and SUL-TMP resistance seems to be similar in human and animal diagnostic isolates. Please verify.

L204. Are the OS-MRSP related? Which ST? same origin?

L219. Were these *S. pseudintermedius* MRSP? Do they belong to the same ST and which one? Were the patients hospitalized. Any evidence of possible nosocomial acquisition?

L234. This is not a pathogenicity island. Please remove throughout the entire manuscript. Also better name the element, which is a known transposon (see comment for Figure 3).

L237. Which element contains *dfrG* and *erm(B)*? Other elements containing resistance genes, including SCCmec, should be characterized.

L241. Are the 17 isolates related? Which ST?

L242. Short read technology may not be able to determine if the others are also located next to the SCCmec elements.

L243. This is not a transposon, but the *mec* gene complex within the SCCmec element. Which IS is located next to the *mecA*? IS 1272? Which type of SCCmec elements?

L247. Are prophages present in the different genomes? If yes, this cannot be said.

L252. Also here: are prophages present in the genomes of household strains?

L310. Did you find any mutations in isoleucyl-tRNA synthetase (IRS) which can be associated with mupirocin resistance?

L331. Association of *murG* with reduced susceptibility to antibiotics is an observation. This should be clearly stated, since association of mutations in *murG* with resistance has not been demonstrated experimentally. Additionally, do the strains with reduced susceptibility to oxacillin and SMX-TMP contain *mecA*, *dfr* genes or mutations in *folA* and *folP* genes? This would explain decrease susceptibility to these antibiotics.

L347. Is the number of related strains higher in MRSP than in MSSP?

L349. Remove terminology of "pathogenicity island". These elements do not contain virulence factors per se, and the use of "pathogenicity island" may lead to confusion.

L351. Any mutations associated with ciprofloxacin resistance?

L356. Delete Interestingly. This is known that clinical strains are more likely to be resistant to antibiotics.

L364. Any prophages in the genomes?

L381-L384. See comment L331.

L386. Is the diversity observed by MLST confirmed by whole genome analysis?

L411-L413. The presence of these genes is well known among MRSP and has been described in several studies. Please either rephrase or delete.

L485. Why mentioning *S. delphini* here if only *S. pseudintermedius* strains were analyzed?

L621. Supplementary Files 2 to 4 were not available in the pdf.

Figure 2. Better differentiate MRSP and MSSP. Include the ST for each strain below the figure next to the strain names. Add mutations for ciprofloxacin and mupirocin resistance if any?

Figure 3.

1) D. This is not a pathogenicity island, but a Tn5405-like transposon containing resistance genes. Linkage of cluster *aph(3)-IIIa-sat4-ant(6)-Ia* with *ermB* is also well known. Please use the existing nomenclature to describe the genetic elements containing resistance genes.

2) CAT, this is the abbreviation for the mechanism. Genes should be in italic and in capital letter. Which gene is responsible for chloramphenicol resistance?

3) Red brackets contain resistance genes, not virulence genes. Please correct. Additionally, not all resistance genes detected are listed here, and *mecA* is not part of the group in the red bracket, but above. Why? Please better group the different genes based on their respective functions.

4) add STs

Reviewer #3 (Remarks to the Author):

The authors performed a core genome analysis on a large set of *S. pseudintermedius* isolates from the United States, covering human and animal isolates differentiated in clinical or colonizing isolates. As the authors did not differentiate between methicillin-susceptible and methicillin-resistant *S.*

pseudintermedius isolates, the results are sometimes misinterpreted, and I have some major and minor comments that must be addressed.

General major comments

The abstract describes that 572 isolates were sequenced. After filtering (line 129), 501 assemblies retained, of which 493 *S. pseudintermedius* genomes. However, bioproject PRJNA908872 contains 236 *S. pseudintermedius* genome assemblies, not 493. All data should be uploaded.

A large part of the bioinformatic analysis was based on the core gene content generated with the tool Roary. However, this tool aligns genes that are present in all genomes and does not comprise the whole genome as no intergenic regions are included. Therefore, the analysis is not performed with a core genome as described throughout the whole manuscript, but this is a core gene analysis. This should be corrected throughout the whole manuscript (e.g. lines 42-43, 153, 172, 182, 191, 194, 211, 213, 229, 282, 341, 478, 485, 523, 525, 532, Figure 1A, Supplemental Figure 5, etc).

The with Roary generated core gene content is small compared to the whole genome. If the content of 1673 core genes is used, which is merely ca 20% of the pangenome according to figure 1D, this is a core content of 60-70% of the complete genome, which is very low for a core content of *S. pseudintermedius* genomes. For example, a core genome analysis with *S. pseudintermedius* genomes (both MRSP and MSSP) performed with parsnp of the Harvest suite (<https://doi.org/10.1186/s13059-014-0524-x>) generates a core genome of 2.2 Mbp, which is ca 75-90% of the total genome.

The Roary core gene analysis of only core genes is not taking intergenic regions into account and is not reliable and robust enough to perform a thorough core genome analysis. Moreover, the identified clusters and SNPs, were described as “whole genome analysis” of “genome-wide analysis”, which is not the appropriate naming of the applied core genome analysis.

The Roary core gene alignment analysis in Figure 1A describes isolate-specific SNPs with Snippy performed on the Roary output (lines 523-526) and Supplemental Figure 3A. These should be replaced with an analysis and alignment that generates a real core genome content, for example with parsnp of the Harvest suite (<https://doi.org/10.1186/s13059-014-0524-x>)

In lines 154 -169 is the population structure and the gene content in relation with host association. The authors found the high genetic diversity remarkable but that is complete in line with the diversity observation in the methicillin susceptible MSSP population described by Haenni et al., <http://dx.doi.org/10.1016/j.jgar.2020.02.016> and Wegener et al., <https://doi.org/10.3390/antibiotics10070854>. Correct me if I am wrong, I have counted 43 MRSP of 493

isolates, so most of the analyzed isolates in this study are MSSP. The section must be rewritten and, in the context, what is known from other studies on the population of MSSP. Lines 168-169 must be deleted this is not correct. In Fig 1A, are only the novel MLST types shown, but all MLST types must be shown, especially the MRSP ones as they are often clonal.

In lines 193-204, the antimicrobial resistance gene content is described, but it is already known that the number of antimicrobial resistance genes can be higher in MRSP but also in methicillin (oxacillin) susceptible *S. pseudintermedius* (MSSP) from human infections. This section also requires more data on breakpoint levels, as among MRSP oxacillin resistance levels vary, and it is remarkable that a high number of strains (11/43) carry the *mecA* gene but shown no oxacillin-resistant phenotype. This must be checked and explained.

All the figures in Fig 2 are difficult to follow as it is unclear how many strains were included in the analysis. As not for all genomes phenotypic antimicrobial resistance data are available (suppl. Table 2, sheet resistance N/A) and for all human diagnostic genomes the resistance is deduced by interpretation with resistant or susceptible. Because it was concluded that the human diagnostic samples have the highest resistance prevalence, but this is not verifiable with the AST data. Furthermore, the genomic data of the identified ARGs are not shown in tables, all detected ARGs per genome should be added to the supplemental material. It is unclear how many strains were included and how reliable the statistics are when low number are used. This especially accounts for MRSP that are around 40 isolates. This section requires detailed information on breakpoints, resistance genes and revised statistics, and OS-MRSP should be labeled as MSSP.

In Figure 3D and 3E the pathogenicity Island and CRISPR-cas system are presented, but short-read assemblies were used it must be explained how the CRISPR-cas operons were concatenated (line 543) assuming that not all operons were one contig. These repetitive elements generally cause contig breaks and are misassembled. So how can you be sure about the gene order and completeness of the operons after concatenation? This question applies also for the pathogenicity island of Figure 3D, again also assuming that this was not on one contigs due to the IS elements found in this island. How were these concatenated and how reliable is this concatenation as these elements contain multicopy sequence repeats that can be dispersed in the genome? Please describe/clarify this in the manuscript.

Minor remarks

Lines 211-214; The cluster/lineage assignments should be included to see which genomes belonged to the same cluster/lineage, for example add this information to the supplemental material. And is there a rationale behind the chosen 25 and 600 SNPs as cut-off for the assignments? As mentioned before, Roary is not suitable for this kind of analysis, and SNPs from the core genome (not only genes but also the intergenic regions) should be used.

Lines 233-235/240-242; as mentioned before identified ARGs should be included in the supplementary table, as well as the cas genes.

Lines 245-247; Figure 3C is about clusters and the text above this sentence is mainly about clusters. How did the authors observe this in lineages?

Lines 249-279; where all CRISPR systems assembled in one contig? For the systems that were not on one contig, these cannot be included in the analysis, since parts/repeats/spacers can be missing.

Line 264, 270; in the previous paragraph, lineages and clusters were defined. In these lines, lineage clusters are described. Please explain or rephrase, are these clusters, are these lineages, or is this a new defined assignment of a certain group?

Line 285; please explain this, where is this information described?

Lines 286-287; as mentioned before, please add the information about which genome belongs to which clusters, for example in the supplemental table.

Lines 302-307 belongs to material and methods.

Line 311: which representative *S. pseudintermedius* genome is used?

Line 343-344: should be rephrased, see the comment on other publication that have shown the genetic diversity

Line 479; which parameters are used for Unicycler?

Line 486; which parameters are used for Roary?

Line 514; “)” is missing at the end of the sentence.

Line 543: please explain how the operons were concatenated if they were located on different contigs. How can you be sure about the composition and completeness after concatenation?

Supplemental Figure 1 – materials and methods; which selective media is used?

Figure 1B-1C-1E; does not show any significant clustering or score -> can be removed to supplementary content.

Figure 3A; What does this represent, are all genomes included here? How many lineages and clusters are defined. I would suggest deleting this figure, since it does not contribute to anything.

Figure 3B; check caption, is it total isolates in a strain? And does this mean that 10 clusters are defined? Why are only clusters shown, and not lineages? Is this necessary information, or can this figure be deleted.

Figure 3C; “only isolates or strain clusters with at least 1 GWAS-identified gene are displayed”, means that there were isolates without any genes? Please rephrase.

Figure 5A: very difficult to read the numbers, increase resolution.

REVIEWER COMMENTS

Reviewer #1 (Remarks to the Author):

Sawhney et al. present their genomic investigation of 493 *S. pseudintermedius* isolates sampled from animal and human clinical sources, colonized human and animals, and household fomites. The authors report gene acquisition (in diagnostic isolates) and mutation (in household isolates) act to differentiate isolates by niche type. Overall, this well-written manuscript presents interesting new findings regarding mechanisms (CRISPR systems and nonsynonymous mutations in defense genes) that contribute to niche differentiation and adaptation in co-circulating bacterial lineages. The methodology is sound and reproducible. The conclusions are supported by the data. My comments are listed below.

1. Font size in all figures was too small to be legible.

We apologize for this error and have updated the font sizes in figures for improved legibility. We will reconfirm with the Editors that our main text figures and supplementary files are acceptable.

2. Although most pet isolates in the study came from dogs, a few were obtained from cats. Dogs and cats are also unique ecological niches that may also shape the population structure and differentiation of *S. pseudintermedius*. This must also be acknowledged and/or recognized as a limitation.

We thank the Reviewer for this comment and recognize the companion animal species would likely represent a unique ecological niche that would influence the population architecture of *S. pseudintermedius*. For this reason we have added a clarifying statement in the discussion (lines 479-482) acknowledging this limitation.

3. The rationale for using 25 whole genome SNPs (line 211) and 600 core genome SNPs (line 213) to define clusters or lineages need to be stated.

We thank the Reviewer for allowing us to expand on this method. 600 core gene SNPs was empirically chosen as the cutoff for lineage definitions based on the histogram in the inlet of Supplementary Figure 6A. This "lineage" definition is only used to preliminarily describe and bin closely related isolates, which were then rigorously compared across the entire length of each genome ($\geq 98\%$ genome coverage was required for strain-level assessment). With this stringent coverage cutoff met, 25 SNPs was chosen as it corresponds to 99.999% ANI, which is typically reported as the whole genome SNP cutoff for strains (PMID 33462508). Lines 635-637 in the Methods section and 256-260 in the Results section have been lightly edited for additional clarity.

4. Avoid starting a sentence with numerical figure (e.g., lines 285, 310, 361). For example, instead of "5 of the 91 NSS genes..." (line 310), you can say "A total of 5 of the 91 NSS genes ..." or something similar.

We thank the Reviewer for highlighting this grammatical error. We have revised all instances in which a sentence began with a numeric symbol.

5. Line 514: Missing close parenthesis

We thank the Reviewer for catching this grammatical error and have corrected it.

6. Intro, lines 65-66 "methicillin resistance – which is very closely tied to multidrug resistance": Missing citation

We appreciate the clarification requested and have specified the reference we are citing directly after the statement.

7. Intro, line 112 "whole genome architecture does not track with host-species or niche": The use of track in this sentence is confusing.

We thank the Reviewer for this comment. We have entirely removed this sentence in response to another Reviewer's request to avoid summarizing Results in the Introduction.

8. Supplementary table 1 should also include genomic features of each isolate, including N50, number of contigs, number of protein coding genes, % completeness, % contamination, genome length, date of isolate collection, body source or disease, accession number

It has come to our attention that some of our submitted Supplementary Files regrettably did not make it to the Reviewers. We apologize for the inconvenience and will confirm with the Editor that Reviewers receive all relevant files moving forward. BioSample identifier, N50, contig count, completeness %, contamination %, and total length are available in Supplementary File 3. Date of isolate collection and body source are available in Supplementary File 1.

9. Supplementary figure 4: legend should indicate what HI and SF represent.

We apologize for this oversight and have defined HI and SF in the legend.

10. It is unclear why *S. delphini*, *S. intermedius* and *S. ursi* were included in the analysis.

We appreciate the clarification requested from the Reviewer. We have written in our intro that SIG consists of closely related *Staphylococcus* species that includes *S.pseudintermedius*, *S. intermedius*, *S. delphini*, and *S. ursi*. While our MALDI-TOF identification system is highly reliable, we endeavored to confirm species identification via average nucleotide identity-based Haddamard matrix (Supplementary Figure 2) and contextualize our isolates within the larger collection of SIG species.

11. Methods, line 431: How the diagnostic canine and feline SIG isolates were selected from the larger submission collection must be stated. It was also no clear whether each isolate included in the study represented a single individual, or were there instances of multiple isolates from the same human or animal? In either case, this should be clarified.

We apologize for the omission. SIG isolates were selected from canine and feline diagnostic submissions to the Kansas State Veterinary Diagnostic Laboratory. Isolates were identified from a frozen isolate collection sequentially starting on January 1 of each calendar year (2016, 2017, 2018 and 2019). Isolates recovered from non-descript specimen types ("swab"), duplicate submissions from the same veterinarian-owner combination (over the entire study period) and non-viable isolates were not eligible for inclusion in this study and were replaced with the next eligible isolate until a total of 25 isolates were identified per calendar year. This clarifying text has been added to the manuscript at the referenced line.

We thank the Reviewer for this comment and have updated Supplementary File 1 with a Patient ID column.

Reviewer #2 (Remarks to the Author):

NCOMMS-23-05743

The authors sequenced 672 strains of *Staphylococcus* of the *intermedius* group (SIG) using Illumina. After quality filtering of the sequence reads, 501 were retained as *Staphylococcus pseudintermedius* for further analysis. The strains were of different human, animal and environment origins and were classified as household isolates and diagnostic isolates. The authors performed WGS-based phylogenetic and comparative analysis as well as antibiotic resistance gene detection and CRISPR identification to determine if differences exist between the isolates from the different origins and niches. Additionally, they analyzed a set of isolates which were isolated from households which were part of a *S. aureus* decolonization program. This study generated a large amount of data and gives new insights into the genomics and genetic diversity of *S. pseudintermedius*. Genomic analysis has been well performed. However, analysis of the resistance mechanisms and associated mobile genetic elements needs to be improved. Presence of virulence factors and prophages should be investigated to support the conclusion.

1. L39. Replace 572 with the actual number of *S. pseudintermedius* used for analysis (n=501)(see L129). Otherwise, it is confusing.

We thank the Reviewer for this clarification request and have replaced 572 with 501.

2. L45. CRISPR systems were also present in clinical isolates, mainly associated with SCCmec elements. Why only mentioning the household isolates here?

We thank the Reviewer for this comment. The Reviewer is correct that CRISPR elements are also present in diagnostic isolates; however, we clarify that the niche-dependent spacer sharing observation is only present among Household isolates.

3. L46. Genes associated with resistance to oxacillin and trimethoprim. This cannot be said as long as it has not been proven experimentally. This is a speculation and should be deleted from the abstract.

We appreciate the Reviewer's comment and agree that our research group has not confirmed such genes confer both oxacillin and trimethoprim resistance among our isolates, and state in the Results section only that such functions have previously been linked to these genes and published upon. Per reviewer suggestion we have rewritten the sentence in the Abstract to decouple the mentions of the observed mutations from the reductions in drug susceptibility.

4. L67. "methicillin-resistant"

We thank the Reviewer for catching this grammatical error and have corrected it.

5. L78. Define MSSP. The authors should also clearly differentiate their isolates as MSSP and MRSP in the study. These specifications would help to better distinguish between antibiotic-susceptible and -resistant isolates.

We thank the Reviewer for requesting this clarification and have defined MSSP here. We appreciate the emphasis on MRSP/MSSP and have defined isolates by their status, as well as strain type, as appropriate throughout the manuscript.

6. L111-L118. This part is a summary of the results and should be removed from the Introduction.

We thank the Reviewer for this comment and have removed the last 3 sentences of the Introduction that summarized the Results.

7. L116. What defines a defense gene? Against what?

Genes are classified as “defense genes” if they are annotated as such using the Cluster of Orthologous Genes (COG) definition. COG had been defined at first mention in the text, though that sentence has now been removed as advised in the above comment.

8. L122. "staphylococci"

We thank the Reviewer for catching this error. It has been corrected.

9. L122-L138. This part consists mainly of M&M and should be shortened. Only results should be presented.

We thank the Reviewer for this suggestion and have removed content that is already stated in the Methods section.

10. L136-L138. This sentence is already mentioned in the M&M section. Avoid repetition.

We thank the Reviewer for this suggestion and have removed content that is already stated in the Methods section.

11. L154. Delete "surprising". It is well known that the diversity is higher among MSSP.

We thank the Reviewer for this comment. The sentence header has been amended to remove surprise at the high diversity observed, and now includes references to studies that have previously reported this (PMIDs 32097758, 34356775).

12. L157-L165. This paragraph should better differentiate between the STs which have been found in MSSP and the STs of MRSP. It should also be mentioned if more diverse STs were found in MSSP than in MRSP. Which STs are associated with MRSP. Are they known? Is ST181 the only ST associated with MRSP in this study?

We thank the Reviewer for this suggestion and have added clarifying text specifying which of our most commonly captured STs encompass MSSP and which encompass MRSP. We also add text on our study as the first observation of MRSP within ST862.

13. L174. Delete surprisingly.

We have removed “surprisingly” from this sentence.

14. L174. Any link between clusters and ST?

Isolates do appear to align by ST and lineage cluster; however, we do not have large enough sample sizes for most STs to conduct rigorous statistical testing.

15. L188. Define COG

We apologize for this oversight and have now defined COG as Cluster of Orthologous Genes here.

16. L193. Here again: Are the isolates belonging to these clades MRSP? Do they belong to specific STs?

We appreciate the Reviewer's comment and have added a ring to Figure 1A to denote *mecA* presence, allowing the reader to more easily link MRSP status with ST. We have also added text stating that MRSP ST181 is represented within these clades.

17. L200. Add fluoroquinolone phenotype and genotype in Figure 2. Do the veterinary diagnostic lab test for ciprofloxacin? Please check if the strains have mutations in the quinolone resistance-determining region of *gyrA* and *parC*.

No fluoroquinolone-conferring resistance genes were identified by the AMRFinder tool. Susceptibility to ciprofloxacin, enrofloxacin, and delafloxacin was phenotypically assessed for most animal diagnostic and household isolates and are reported in Supplementary File 2.

18. L201. Based on Figure 2, the abundance of *mecA* gene and SUL-TMP resistance seems to be similar in human and animal diagnostic isolates. Please verify.

The authors appreciate the Reviewer's comment for clarification. We have verified that Figure 2B was missing human diagnostic information for TMX. We have corrected this information in the manuscript text and Figure 2. Our original results that diagnostic isolates have significantly more phenotypic resistance against oxacillin, doxycycline, and ciprofloxacin remains true. As suspected, there is no significant differences amongst animal diagnostic nor human diagnostic resistant phenotypes for TMX when an FDR-corrected chi-squared analysis was performed on all sample types for TMX interpretations.

19. L204. Are the OS-MRSP related? Which ST? same origin?

The 10 OS-MRSP isolates are phylogenetically diverse, together representing 9 sequence types: HI_0003 (ST181), HI_0014 (ST64), HI_0075 (ST2484), HI_0088 (ST261), HI_0186 (ST551), AI_0026 (ST311), AI_0035 (ST2408), AI_0053 (ST64), AI_0068 (ST555), SF_0060 (ST862). Text has been added to this line for clarification.

20. L219. Were these *S. pseudintermedius* MRSP? Do they belong to the same ST and which one? Were the patients hospitalized. Any evidence of possible nosocomial acquisition?

We thank the Reviewer for allowing us to expand on this observation. All but one of these strain clusters are MSSP (strain cluster 51 representing HI_0112 and HI_0113 – both ST2503 – is MRSP). We do not have access to metadata on inpatient vs. outpatient status but do report the site of body site of collection below; in sum, they vary and include tissue, wound, respiratory, abscess, and drainage. For the strain clusters representing isolates from different patients, these isolates are collected 2 – 212 days apart (median: 25 days); three of these strain clusters represent isolates collected 2, 8, and 9 days apart. Based on these data we can speculate that in some cases acquisition of the same strains through strain persistence in the built environment or transmission via co-colonized hospital staff may have occurred. We also observe one patient harboring isolates of 2 different strain clusters, each of which were found in multiple patients within the hospital system (bolded below). We have added text to this line to expand on this observation.

The data below include strain cluster #, MSSP/MRSP status, isolates within strain cluster, description of patients these isolates are collected from, in parentheses (age, sex, body site, date of collection).

19 (all MSSP): HI_0164, 0165, 0166 – **same** patient at same time (62.2 F tissue, 4/2/19)
23 (all MSSP): HI_0021, 0022 – **same** patient at same time (54.2 F wound, 8/24/15)
24 (all MSSP): HI_0083, 0096 – different patients, 3 mo apart (35.1 F wound 62220 11/7/17; 6.5 M respiratory 65401 2/8/18)
27 (all MSSP): HI_0124, 0128, 0131 – **same** patient across 1.5 mo (50.3 F respiratory 62711 8/21/18 – 10/8/18)
42 (all MSSP): HI_0025, 0026 – different patients, 1 week apart (37.1 F abscess 10/25/15; 58.3 F wound 62034 11/3/15)
43 (all MSSP): HI_0050, 0054 – **same** patient across 1.5 mo (61.5 F wound 62010 1/24/17 – 3/13/17)
44 (all MSSP): HI_0056, 0058 – different patients across 3 wks (69.8 M wound 63017 5/12/17; **69.3 M respiratory 63129 6/6/17**)
45 (all MSSP): HI_0059, 0060 – different patients across 2 days (**69.3 M respiratory 63129 6/6/17**; 62 M tissue 62232 6/8/17)
46 (all MSSP): HI_0073, 0074 – different patients across 1 wks (40.4 F drainage 63121 9/5/17; 88 M wound 62087 9/13/17)
48 (all MSSP): HI_0077, 0078 – **same** patient across 1 day and different sites (70.2 M 62271 wound 10/4/17; tissue 10/5/17)
49 (all MSSP): HI_0094, 0097 – different patients across 1 mo (76.3 M other 62010 2/9/18; 66.6 M wound 63033 3/2/18)
51 (all **MRSP**): HI_0112, 0113 – **same** patient same time (55.3 M wound 63136 7/1/18)
52 (all MSSP): HI_0123, 0135 – different patients across 2 mo (38.2 F wound 63620 8/17/18; 57.4 M abscess 62012 11/10/18)
53 (all MSSP): HI_0141, 0152 – **same** patient across 1.5 mo (63.6 M respiratory 63127 12/10/18 – 1/25/19)
55 (all MSSP): HI_0143, 0187 – different patients across 7 mo (2.6 F drainage 62034 12/14/18 – 79.1 F wound 63129 7/14/19)
56 (all MSSP): HI_0171, 0188 – different patients across 3 mo (86 M tissue 62012 5/1/19; 3.2 F drainage 62034 7/26/19)

21. L234. This is not a pathogenicity island. Please remove throughout the entire manuscript. Also better name the element, which is a known transposon (see comment for Figure 3).

We thank the Reviewer for correcting this error. We have replaced all instances of "pathogenicity island" with "resistance gene cluster".

22. L237. Which element contains *dfrG* and *erm(B)*? Other elements containing resistance genes, including SCCmec, should be characterized.

When present, resistance genes *dfrG* and *erm(B)* were either not a member of a transposable element or were adjacent to genes that could only be annotated nonspecifically by PGAP as "insertion element protein". Our intention for Figures 3D-E are to visualize the two large syntenic gene clusters (resistance gene cluster and CRISPR-Cas operon) identified by the pan-GWAS to be enriched among diagnostic isolates; while *mecA* is also enriched, it is not a member of either cluster.

23. L241. Are the 17 isolates related? Which ST?

These 17 isolates are represented by 11 different STs: ST 1229 (n=3), ST 181 (n=3), ST 1055 (n=2), ST 68 (n=2), ST 496 (n=1), ST 2392 (n=1), ST 121 (n=1), ST 155 (n=1), ST 2275 (n=1), ST 2455 (n=1), and ST 551 (n=1). Text has been added to this line for clarification.

24. L242. Short read technology may not be able to determined if the others are also located next to the SCCmec elements.

We thank the Reviewer for this comment and agree that the two genetic elements may be syntenic in more than the 3 mentioned isolates; however, given the limitations of short-read sequencing, we can only state synteny with confidence for these 3 isolates.

25. L243. This is not a transposon, but the *mec* gene complex within the SCCmec element. Which IS is located next the *mecA*? IS 1272? Which type of SCCmec elements?

We thank the Reviewer for flagging this error and have corrected the text to refer to the gene cassette as the SCCmec element. The PGAP annotation tool described the IS solely as "IS6 family transposase".

26. L247. Are prophages present in the different genomes? If yes, this cannot be said.

We thank the Reviewer for raising this concern. Given the proximity of the Cas proteins alongside the CRISPR spacers, we believe we are identifying CRISPR-Cas systems and not prophages. Still, we have amended the text in lines 316-319 in consideration of prophage presence.

27. L252. Also here: are prophages present in the genomes of household strains?

Addressed in above response to #26.

28. L310. Did you find any mutations in isoleucyl-tRNA synthetase (IRS) which can be associated with mupirocin resistance?

We thank the Reviewer for raising this question. We did not observe any mutations in isoleucyl-tRNA synthetase among the persisting household strain clusters.

29. L331. Association of *murG* with reduced susceptibility to antibiotics is an observation. This should be clearly stated, since association of mutations in *murG* with resistance has not been demonstrated experimentally. Additionally, do the strains with reduced susceptibility to oxacillin and SMX-TMP contain *mecA*, *dfr* genes or mutations in *folA* and *folP* genes? This would explain decrease susceptibility to these antibiotics.

We thank the Reviewer for this comment and concur that our research group has not experimentally validated the *murG* mutation for reduced susceptibility. We have added “we observed” in this sentence for clarification and further caution against overinterpretation in the Limitations paragraph in the Discussion.

The strain cluster 03 isolates with reduced susceptibility to OXA and SMX-TMP only harbor *blaZ* and do not harbor mutations in any of the genes listed.

30. L347. Is the number of related strains higher in MRSP than in MSSP?

There are fewer unique STs represented by MRSP than MSSP, reflecting the far greater numbers of MSSP isolates captured in our cohorts.

31. L349. Remove terminology of "pathogenicity island". These elements do not contain virulence factors per se, and the use of "pathogenicity island" may lead to confusion.

We thank the Reviewer for correcting this error. We have replaced all instances of "pathogenicity island" with “resistance gene cluster”.

32. L351. Any mutations associated with ciprofloxacin resistance?

Addressed in above response to #17.

33. L356. Delete Interestingly. This is known that clinical strains are more likely to be resistant to antibiotics.

“Interestingly” has been deleted.

34. L364. Any prophages in the genomes?

Addressed in above response to #26.

35. L381-L384. See comment L331.

Addressed in above response to #29.

36. L386. Is the diversity observed by MLST confirmed by whole genome analysis?

Yes, isolates of the same strain cluster share the same ST.

37. L411-L413. The presence of these genes is well known among MRSP and has been described in several studies. Please either rephrase or delete.

We concur that the presence of these genes is well known among MRSP. Our point is to highlight that these genes are overrepresented among isolates found in diagnostic settings relative to isolates found in commensal settings.

38. L485. Why mentioning *S. delphini* here if only *S. pseudintermedius* strains were analyzed?

We also create a maximum-likelihood phylogenetic tree for the *S. delphini* isolates in our cohort, which utilize these computational tools and is visualized in Supplementary Figure 2B.

39. L621. Supplementary Files 2 to 4 were not available in the pdf.

We apologize for this error and will confirm with the Editor that all files are included and available for review.

40. Figure 2. Better differentiate MRSP and MSSP. Include the ST for each strain below the figure next to the strain names. Add mutations for ciprofloxacin and mupirocin resistance if any?

The analysis described in Figure 2A is limited to assessment for the presence of entire resistance-conferring genetic elements as annotated by AMRFinder. We do not assess for mutations within genes conferring resistance here. We recognize the importance of MRSP/MSSP status, which is presently differentiated through the *mecA* and Oxacillin rows within Figure 2A. ST has been added for each isolate represented in Figure 2A.

41. Figure 3.

- a. D. This is not a pathogenicity island, but a Tn5405-like transposon containing resistance genes. Linkage of cluster *aph(3)-IIIa-sat4-ant(6)-Ia* with *ermB* is also well known. Please use the existing nomenclature to describe the genetic elements containing resistance genes.

Figure legend has been revised to include “Tn5405-mediated resistance gene cluster” in place of “pathogenicity island.”

- b. CAT, this is the abbreviation for the mechanism. Genes should be in italic and in capital letter. Which gene is responsible for chloramphenicol resistance?

We thank the Reviewer for flagging this. The gene is *catA* and the panel has been updated to reflect this.

- c. Red brackets contain resistance genes, not virulence genes. Please correct. Additionally, not all resistance genes detected are listed here, and *mecA* is not part of the group in the red bracket, but above. Why? Please better group the different genes based on their respective functions.

Figure legend has been revised to replace “pathogenicity island” with “resistance gene cluster.” Gene *mecA* and other resistance genes are not part of the red bracketed group because they are not found within the resistance gene cluster. The grouping of genes (rows in the heatmap in Figure 3C) is determined by clustering isolates and strain representatives by similar gene presence/absence profiles – genes listed above and below each other in the heatmap do not necessarily mean they are syntenic.

- d. add STs

We thank the Reviewer for requesting this edit and have added ST annotation to this panel.

Reviewer #3 (Remarks to the Author):

The authors performed a core genome analysis on a large set of *S. pseudintermedius* isolates from the United States, covering human and animal isolates differentiated in clinical or colonizing isolates. As the authors did not differentiate between methicillin-susceptible and methicillin-resistant *S. pseudintermedius* isolates, the results are sometimes misinterpreted, and I have some major and minor comments that must be addressed.

General major comments

1. The abstract describes that 572 isolates were sequenced. After filtering (line 129), 501 assemblies retained, of which 493 *S. pseudintermedius* genomes. However, bioproject PRJNA908872 contains 236 *S. pseudintermedius* genome assemblies, not 493. All data should be uploaded.

We thank the Reviewer for flagging this and apologize for the oversight. All isolate short reads and assemblies are now uploaded to NCBI under PRJNA908872.

2. A large part of the bioinformatic analysis was based on the core gene content generated with the tool Roary. However, this tool aligns genes that are present in all genomes and does not comprise the whole genome as no intergenic regions are included. Therefore, the analysis is not performed with a core genome as described throughout the whole manuscript, but this is a core gene analysis. This should be corrected throughout the whole manuscript (e.g. lines 42-43, 153, 172, 182, 191, 194, 211, 213, 229, 282, 341, 478, 485, 523, 525, 532, Figure 1A, Supplemental Figure 5, etc).

We thank the Reviewer for this observation. Where appropriate, "core genome" and "accessory genome" have been replaced with "core gene" and "accessory gene", and "whole genome content" has been replaced with "global encoded gene content". All mentioned lines have been revised, except those pertaining to the pan-GWAS (original lines 229, 532), which reflect language used by the publishers of Scoary to describe their tool's capabilities, and whose pipeline (Prokka-PGAP/Roary/Scoary) we replicated.

3. The with Roary generated core gene content is small compared to the whole genome. If the content of 1673 core genes is used, which is merely ca 20% of the pangenome according to figure 1D, this is a core content of 60-70% of the complete genome, which is very low for a core content of *S. pseudintermedius* genomes. For example, a core genome analysis with *S. pseudintermedius* genomes (both MRSP and MSSP) performed with parsnp of the Harvest suite (<https://doi.org/10.1186/s13059-014-0524-x>) generates a core genome of 2.2 Mbp, which is ca 75-90% of the total genome.

Our core gene alignment is based on 1,704 genes shared among $\geq 99\%$ of isolates in our cohort. The mean total gene count for isolates is 2387.00 genes, meaning our core gene alignment reflects 71.4% of all genes encoded in each isolate's genome. Still, the reviewer raises an important point that our analysis does not include intergenic regions that may also be part of the core genome. As such, we have complemented our analysis with a true core genome alignment via the suggested Harvest suite. These data are shown in the new Supplementary Figure 4. We observe that the core genome tree generated by Parsnp (Harvest) reflects our core gene alignment findings from Roary, in that no clustering is apparent by niche type or host species.

4. The Roary core gene analysis of only core genes is not taking intergenic regions into account and is not reliable and robust enough to perform a thorough core genome analysis. Moreover, the identified clusters and SNPs, were described as "whole genome analysis" of "genome-wide analysis", which is not the appropriate naming of the applied core genome analysis.

Addressed in above response to #2.

5. The Roary core gene alignment analysis in Figure 1A describes isolate-specific SNPs with Snippy performed on the Roary output (lines 523-526) and Supplemental Figure 3A. These should be replaced with an analysis and alignment

that generates a real core genome content, for example with parsnp of the Harvest suite (<https://doi.org/10.1186/s13059-014-0524-x>)

Addressed in above response to #3.

6. In lines 154 -169 is the population structure and the gene content in relation with host association. The authors found the high genetic diversity remarkable but that is complete in line with the diversity observation in the methicillin susceptible MSSP population described by Haenni et al., <http://dx.doi.org/10.1016/j.jgar.2020.02.016> and Wegener et al. <https://doi.org/10.3390/antibiotics10070854>. Correct me if I am wrong, I have counted 43 MRSP of 493 isolates, so most of the analyzed isolates in this study are MSSP. The section must be rewritten and, in the context, what is known from other studies on the population of MSSP. Lines 168-169 must be deleted this is not correct. In Fig 1A, are only the novel MLST types shown, but all MLST types must be shown, especially the MRSP ones as they are often clonal.

We thank the Reviewer for this comment. The sentence header has been amended to remove surprise at the high diversity observed, and now includes citations to both referenced studies. To clarify, Figure 1A lists all MLSTs represented by ≥ 5 isolates, not just those that are novel. We agree that it is important to list the MLST for each isolate, and this information is available in Supplementary File 3. Given that we observe 315 unique MLSTs, it is not feasible to represent all in a color bar in Figure 1A. We appreciate the Reviewer's comment that MRSP isolates are often clonal, and have added a ring to Figure 1A to reflect presence of *mecA*. Original lines 168-169 have been revised to replace "literature" with "genomes"; this claim of "uniqueness" is supported by our observation that the majority of our captured isolates represent MLSTs that have never before been sequenced and added to the PubMLST database.

7. In lines 193-204, the antimicrobial resistance gene content is described, but it is already known that the number of antimicrobial resistance genes can be higher in MRSP but also in methicillin (oxacillin) susceptible *S. pseudintermedius* (MSSP) from human infections. This section also requires more data on breakpoint levels, as among MRSP oxacillin resistance levels vary, and it is remarkable that a high number of strains (11/43) carry the *mecA* gene but shown no oxacillin-resistant phenotype. This must be checked and explained.

We thank the Reviewer for this comment. A reference has been added (PMID 34796561) to indicate greater ARG burden among MRSP has previously been reported. Exact breakpoint values and clinical interpretations are reported for each isolate in our cohort for each tested antibiotic in Supplementary File 2. The 9 OS-MRSP isolates are all missing *mecA* regulators *mecI* and *mecR1*. This has been added to the manuscript text in lines 249-251 and 439-441 as a plausible explanation for the phenotype.

8. All the figures in Fig 2 are difficult to follow as it is unclear how many strains were included in the analysis. As not for all genomes phenotypic antimicrobial resistance data are available (suppl. Table 2, sheet resistance N/A) and for all human diagnostic genomes the resistance is deduced by interpretation with resistant or susceptible. Because it was concluded that the human diagnostic samples have the highest resistance prevalence, but this is not verifiable with the AST data. Furthermore, the genomic data of the identified ARGs are not shown in tables, all detected ARGs per genome should be added to the supplemental material. It is unclear how many strains were included and how reliable the statistics are when low number are used. This especially accounts for MRSP that are around 40 isolates. This section requires detailed information on breakpoints, resistance genes and revised statistics, and OS-MRSP should be labeled as MSSP.

We thank the Reviewer for highlighting this concern. Figure 2A legend mentions that only isolates with ≥ 2 ARGs are shown. We have updated this sentence to add "(n=188)" for absolute clarity. For Figures 2C and 2D, the full cohort of sequenced *S. pseudintermedius* isolates (n=493) were included. We have updated the figure legend to reflect this. We have also updated the legend in Figure 2D from describing isolates as MSSP/MRSP to *mec*-neg/*mec*-pos. We have now further added both ARG count and list of ARGs present per isolate in tabular format to Supplementary File 2.

We have verified that Figure 2B was missing human diagnostic information for TMX. We have corrected this information in lines 230-236 and have updated Figure 2B. To clarify, FDR-corrected chi-squared analyses were performed on all sample types for each antibiotic (Animal diagnostic n=100, Animal colonizing n=108, Human

diagnostic n=109, Human colonizing=38, Environmental=144). Our original results that diagnostic isolates are significantly more likely to be phenotypically resistant against oxacillin, doxycycline, and ciprofloxacin than household isolates remain true (there are no significant differences between resistance phenotypes for TMX across cohorts).

9. In Figure 3D and 3E the pathogenicity Island and CRISPR-cas system are presented, but short-read assemblies were used it must be explained how the CRISPR-cas operons were concatenated (line 543) assuming that not all operons were one contig. These repetitive elements generally cause contig breaks and are misassembled. So how can you be sure about the gene order and completeness of the operons after concatenation? This question applies also for the pathogenicity island of Figure 3D, again also assuming that this was not on one contigs due to the IS elements found in this island. How were these concatenated and how reliable is this concatenation as these elements contain multicopy sequence repeats that can be dispersed in the genome? Please describe/clarify this in the manuscript.

We thank the Reviewer for this question and agree that transposable elements like the pathogenicity island often break short read assemblies. It is precisely for this reason that we ensured accurate representation of both complete and broken assemblies among the representative isolates shown in Figures 3D-E. Here, the solid black horizontal line represents genes found on the same contig in the exact order they appear, with gaps in the black horizontal line signifying multiple contigs, as is seen in HI_0018, AI_0010, and HI_0186 in Figure 3D. We make no claim to synteny across contig breaks in these isolates, but do show high pairwise ANI between these isolates and the top HI_0093, whose *ant(6)-la-sat4-aph(3')* resistance gene cluster (previously “pathogenicity island”) and upstream regions were successfully reassembled without contig breaks. Interestingly, we did not observe any assembly fragmentation within CRISPR Cas operons in our cohort. In the referenced line 543 (“Contigs harboring CRISPR-Cas operons were extracted from isolate assemblies, concatenated, and inputted into CrisprCasFinder”), here “concatenating” means creating an isolate-specific multifasta file of only the full length contigs with at least one PGAP-annotated Cas gene for bioinformatic analysis. Within this multifasta file, Cas genes and their associated operon remain separate, and tied to their contig of origin. The purpose of the multifasta is only to tie the downstream CRISPR spacers (identified by CrisprCasFinder) to the isolate of origin (the contig of origin is retained). To add further context, as shown in Supplementary File 3, 213 of the 238 Cas operon-positive isolates (89.5%) carry just one operon (meaning only one contig per isolate was found harboring a Cas protein). For most of the remaining Cas operon-positive isolates with multiple Cas operons (1 operon per contig), the second operon was a duplicate of the first and removed. In 11 of these cases, the operons were unique, and thus each was included in the analysis. These are referred to with a suffix “a” and “b” (i.e., HI_0001a and HI_0001b; HI_clstr22a and HI_clstr22b). Clarifying text has been added to lines 653-679 in the “CRISPR Spacer Investigation” within the Methods section. We have also added our full workflow to the github URL listed in the newly-added “Code availability” section (URL: github.com/sanjsawhney/staphylococcus-pseudintermedius; file: “CRISPR Spacer Pipeline.pdf”).

Minor remarks

10. Lines 211-214; The cluster/lineage assignments should be included to see which genomes belonged to the same cluster/lineage, for example add this information to the supplemental material. And is there a rationale behind the chosen 25 and 600 SNPs as cut-off for the assignments? As mentioned before, Roary is not suitable for this kind of analysis, and SNPs from the core genome (not only genes but also the intergenic regions) should be used.

All strain and lineage cluster assignments are available in Supplementary File 3. 600 core gene SNPs was empirically chosen as the cutoff for lineage definitions based on the histogram in the inset of Supplementary Figure 6A. This “lineage” definition is only used to preliminarily describe and bin closely related isolates, which were then rigorously compared across the entire length of each genome ($\geq 98\%$ genome coverage was required for strain-level assessment). With this stringent coverage cutoff met, 25 SNPs was chosen as it corresponds to 99.999% ANI, which is typically reported as the whole genome SNP cutoff for strains (PMID 33462508). Lines 635-637 in the Methods section and 256-258 in the Results section have been lightly edited for additional clarity.

11. Lines 233-235/240-242; as mentioned before identified ARGs should be included in the supplementary table, as well as the cas genes.

We thank the Reviewer for this suggestion. All isolate-specific ARG profiles have been added to Supplementary File 2.

12. Lines 245-247; Figure 3C is about clusters and the text above this sentence is mainly about clusters. How did the authors observe this in lineages?

The motivation behind strain correction prior to this analysis is to avoid overrepresentation of accessory gene content from near-identical isolates. Performing lineage correction would incorporate less closely related isolates (those solely < 600 core gene SNPs apart), which would become outside the scope of our objective.

13. Lines 249-279; where all CRISPR systems assembled in one contig? For the systems that were not on one contig, these cannot be included in the analysis, since parts/repeats/spacers can be missing.

We thank the Reviewer for this question and share their concern. All CRISPR systems were indeed assembled on one contig.

14. Line 264, 270; in the previous paragraph, lineages and clusters were defined. In these lines, lineage clusters are described. Please explain or rephrase, are these clusters, are these lineages, or is this a new defined assignment of a certain group?

We apologize for the confusion and have added clarifying text to the referenced section. The analysis in Figures 4B-D are conducted on pairwise comparisons between lineage cluster representatives (and individual isolates that do not fall into any lineage cluster).

15. Line 285; please explain this, where is this information described?

We thank the Reviewer for requesting this clarification. The HOME2 and SHINE studies tracked 44 unique households. Of these, 23 households harbored ≥ 5 *S. pseudintermedius* isolates. One additional household harboring 4 isolates was also included as it harbored a strain cluster present in multiple households. We have updated the Results text at lines 348-349, the Methods text at lines 689-691, and Figure 5A to better describe this inclusion criteria. Metadata outlining which households harbor which isolates are available in Supplementary Files 1 and 3.

16. Lines 286-287; as mentioned before, please add the information about which genome belongs to which clusters, for example in the supplemental table.

These data are available in the "Cluster" column within Supplementary File 3.

17. Lines 302-307 belongs to material and methods.

We thank the Reviewer for this suggestion and have removed this content from the Results section. The same information is already described in the Methods section under "Within Household Tracking and Evolution".

18. Line 311: which representative *S. pseudintermedius* genome is used?

SF_0321 was used as the reference assembly. This is updated in the figure legend.

19. Line 343-344: should be rephrased, see the comment on other publication that have shown the genetic diversity

We thank the Reviewer for this suggestion and have removed text suggesting such genomic diversity among *S. pseudintermedius* was not previously reported on.

20. Line 479; which parameters are used for Unicycler?

We appreciate identification of this oversight. Unicycler was run at default parameters, which is now clarified in the Methods.

21. Line 486; which parameters are used for Roary?

We appreciate identification of this oversight. Flags have been added.

22. Line 514; “)” is missing at the end of the sentence.

We appreciate identification of this oversight. End-parentheses have been added.

23. Line 543: please explain how the operons were concatenated if they were located on different contigs. How can you be sure about the composition and completeness after concatenation?

Addressed in above response to #9.

24. Supplemental Figure 1 – materials and methods; which selective media is used?

Clinical specimens received for culture-based testing in the Barnes-Jewish Hospital Clinical Microbiology Laboratory were plated to agar medium according to the laboratory’s standard operating procedures for each specimen type. In general, specimens were plated to sheep’s blood, chocolate, and MacConkey agar (Remel, Lenexa, KS). Isolates of *S. pseudintermedius* were most commonly obtained from the sheep’s blood agar plate”.

25. Figure 1B-1C-1E; does not show any significant clustering or score -> can be removed to supplementary content.

We thank the reviewer for this suggestion. We believe it is important to highlight in the main text that clustering by accessory gene content and COG distribution is not observed, since the opposite is observed for *S. pseudintermedius*'s close relative *Staphylococcus aureus* (PMID 30038246).

26. Figure 3A; What does this represent, are all genomes included here? How many lineages and clusters are defined. I would suggest deleting this figure, since it does not contribute to anything.

We apologize for the lack of clarity. This panel only includes isolates that fall into a lineage cluster. Each dot is a pairwise comparison between isolates of the same lineage cluster. The relevance of this panel is to show the distinction between lineage cluster and strain cluster definitions, and to show that many but not all isolates within the same lineage cluster can be considered the same strain.

27. Figure 3B; check caption, is it total isolates in a strain? And does this mean that 10 clusters are defined? Why are only clusters shown, and not lineages? Is this necessary information, or can this figure be deleted.

We apologize for the lack of clarity. The x-axis describes the number of isolates in a strain cluster. The y-axis describes the number of strain clusters with the x-axis defined amount of isolates. Here, we display 56 strain clusters that together describe 184 isolates. The figure legend has been updated accordingly.

28. Figure 3C; “only isolates or strain clusters with at least 1 GWAS-identified gene are displayed”, means that there were isolates without any genes? Please rephrase.

We apologize for the lack of clarity. Only isolates or strain clusters that harbor at least one gene identified by GWAS to be overrepresented within the household or diagnostic cohort are shown.

29. Figure 5A: very difficult to read the numbers, increase resolution.

We thank the Reviewer for highlighting this concern and have increased resolution of Figure 5A as requested.